# Mapping and identification of soft corona proteins at nanoparticles and their impact on cellular association

Hossein Mohammad-Beigi [1,2], Yuya Hayashi [3], Christina Moeslund Zeuthen[1,2], Hoda Eskandari[1,2], Carsten Scavenius [3], Kristian Juul-Madsen[4], Thomas Vorup-Jensen [4], Jan J. Enghild [3] & Duncan S. Sutherland [1,2]✉

The current understanding of the biological identity that nanoparticles may acquire in a given biological milieu is mostly inferred from the hard component of the protein corona (HC). The composition of soft corona (SC) proteins and their biological relevance have remained elusive due to the lack of analytical separation methods. Here, we identify a set of specific corona proteins with weak interactions at silica and polystyrene nanoparticles by using an in situ click-chemistry reaction. We show that these SC proteins are present also in the HC, but are specifically enriched after the capture, suggesting that the main distinction between HC and SC is the differential binding strength of the same proteins. Interestingly, the weakly interacting proteins are revealed as modulators of nanoparticle-cell association mainly through their dynamic nature. We therefore highlight that weak interactions of proteins at nanoparticles should be considered when evaluating nano-bio interfaces.

[1] Interdisciplinary Nanoscience Centre (iNANO), Aarhus University, Gustav Wieds Vej 14, 8000 Aarhus C, Denmark. [2] The Centre for Cellular Signal Patterns (CellPAT), Aarhus University, Gustav Wieds Vej 14, 8000 Aarhus C, Denmark. [3] Department of Molecular Biology and Genetics, Aarhus University, Gustav Wieds Vej 10, 8000 Aarhus C, Denmark. [4] Department of Biomedicine, Faculty of Health, Aarhus University, Høegh-Guldbergs Gade 10, 8000 Aarhus C, Denmark. ✉email: duncan@inano.au.dk

Nanoparticles (NPs) are promising agents for drug delivery and visualization in vivo. Upon exposure to biofluids, the NPs acquire a "protein corona" due to the adherence of host proteins on the NPs surface. The composition of the corona is dependent on the types of nanoparticles and the biological sources[1–3], and considered to provide the NPs with a "biological" identity[4] that affects stability, circulation time, and cellular uptake/interactions, and therefore has a strong impact on the functional role for the NPs[5–10].

The formation of protein corona leads to two main consequences that determine how well the nanoparticles associate with cells. Under a serum-free condition, pristine nanoparticles spontaneously bind to cell membranes in a nonspecific manner lowering their surface energy while protein coronas, in general, reduce this nonspecific interaction as less nanoparticle surface is exposed[11]. In parallel, nanoparticle-bound proteins provide the potential for specific interactions during cell association (CA), including receptor-mediated membrane adhesion and subsequent uptake[12,13], and contribute to the resultant biomolecular corona-defined biological identity of the nanoparticles[14,15]. Therefore, to predict the biological behavior of nanoparticles, it is essential to have a combined understanding of the composition and structure of protein corona.

Kinetic evaluation of protein corona formation and identification of the proteins forming the corona have become active research topics aiming to understand the particokinetics, cellular interactions, and mechanisms of nanoparticle toxicity[16]. In a complex and dynamic process, proteins competitively adhere to the surface of nanoparticles to form a combined "Hard" (HC) and "Soft" corona (SC). HC proteins with a high binding affinity and low dissociation rate remain tightly bound to the surface, whereas SC proteins with a high dissociation rate are rapidly exchanged. At the surface of nanoparticles, proteins can undergo reorientation and conformational changes[17,18], presumably leading to at least a partially denatured state that has a reduced dissociation rate in a process referred to as "hardening"[19]. The evolution and dynamics of HC formation are relatively well studied[14,20–22]; HC is established rapidly, and the evolution of HC over time is only quantitative with altered relative amounts, rather than the changes in protein composition expected from the Vroman effect[12]. The current understanding is that the HC proteins—with their long residence time—give the nanoparticles a biological identity by presenting receptor-binding sites for cellular interactions with a biologically relevant timescale[23]. As SC proteins by definition have a shorter residence time on nanoparticles than HC proteins making them difficult to isolate from free proteins of the mother liquid, their potential biological impacts through specific and/or nonspecific interactions have often been ignored.

Recent work has developed approaches to quantify SC protein binding and address the potential of soft interactions to modulate toxicity by localized sulfidation at the surface of silver nanoparticles[16]. Several methods, such as centrifugation-based separation techniques together with proteomic characterization[24] or multistep centrifugation[25], are proposed to retain a larger fraction of the HC proteins for identification during separation albeit still after long times. In the centrifugation-based separation technique, by using transmission electron microscopy technique, it is shown that protein corona is an undefined loose network of proteins; however, in that method, there is a risk to capture bulk proteins between nanoparticles during centrifugation. Asymmetric flow field-flow fractionation (A4f)[26] and surface plasmon resonance (SPR) coupled with mass spectroscopy[27] have been applied to PEGylated nanoparticles to identify weakly protein-binding proteins in stealth systems. In the later case, SC and HC proteins are identified in a label-free method, and the SC proteins are found as the stealth component of the biological identity. However, it is tested only on liposomes. For the rapidly exchanging SC proteins, several key open questions remain, including whether SC proteins are different from HC proteins, and if there is a role for SC proteins in determining cellular interactions.

To address these critical questions, a comprehensive picture of corona composition and residence time for SC proteins is needed. Here, by developing an experimental approach based on click chemistry, we capture weakly interacting proteins along with HC proteins for mass spectrometry-based compositional profiling, and identify proteins that are either new or with increased amount compared to HC layers as SC proteins. We find that the majority of the identified SC proteins are not unique to SC, but are also present in the HC representing different binding strength states of the same proteins. On the contrary, only a minor fraction of SC proteins are identified exclusively in the SC. Moreover, as our method forces SC proteins to stay in place by cross-linking, such that the SC proteins acquire residence time long enough for biological interactions, we are able to demonstrate a role for the SC proteins in cell association of nanoparticles, that are dependent both on the type of cells and nanoparticles. Therefore, turning off the dynamic nature of dissociation, which is the modulation of real condition for cell studies, provides us the possibility to study the effect of the dynamic nature of SC proteins on cell association.

## Results

**A click-chemistry method captures SC proteins.** Recently, the catalyst-free strain-promoted alkyne azide cycloaddition (SPAAC) "click" chemistry has gained interest in many biological and medical applications due to its high speed, efficiency, specificity, and bioorthogonality[28–31]. Therefore, we have developed a SPAAC click reaction between azide-modified HC proteins on nanoparticles (HC-$N_3$) and dibenzocyclooctyne (DBCO)-activated SC proteins (FBS-D) (Fig. 1a) in order to trap the transiently binding SC proteins on the NPs surface (HC+SC sample). Sulfo-SASD and DBCO-sulfo-NHS were used for the modification of proteins to perform the click-chemistry reaction described in Fig. 1a. The modification occurs through the reaction between sulfo-NHS moieties on the cross-linkers with primary amines on proteins. None of the azide or DBCO-reactive groups on these heterobifunctional cross-linkers react with any of the functional groups on proteins, which avoid cross-linking of HC or SC proteins with other HC or SC ones. Moreover, sulfo-SASD contains a dithiol, which provides a possibility to cleave the covalent bond between proteins by using reducing agents for analysis. Negatively charged hydrophilic silica nanoparticles (SNPs, 70 nm) and hydrophobic carboxyl-modified polystyrene nanoparticles (PsNPs, 100 nm) were used in this study as model nanoparticles[13,32,33]. We used four control samples representing HC and FBS with and without chemical modifications (hard corona (HC), hard corona modified with azide (HC-$N_3$), FBS-D added to HC (D Ctrl), and FBS added to HC-$N_3$ ($N_3$ Ctrl)), and one HC+SC sample that encompasses proteins in both HC and captured SC states.

We first optimized the click-reaction conditions. After formation of HC on SNPs, the concentrations of sulfo-SASD for modification of HC proteins was optimized. The results show that all HC proteins on SNPs were modified with azide at 0.55 mM of sulfo-SASD, with at least one azide group per HC protein (Supplementary Fig. 2). Then, the concentration of DBCO-sulfo-NHS for labeling free FBS proteins was optimized by measuring the degree of labeling and the extent of the click reaction (Supplementary Fig. 3 and Supplementary Table 1). DBCO-modified proteins were then added to SNPs@HC-$N_3$ to

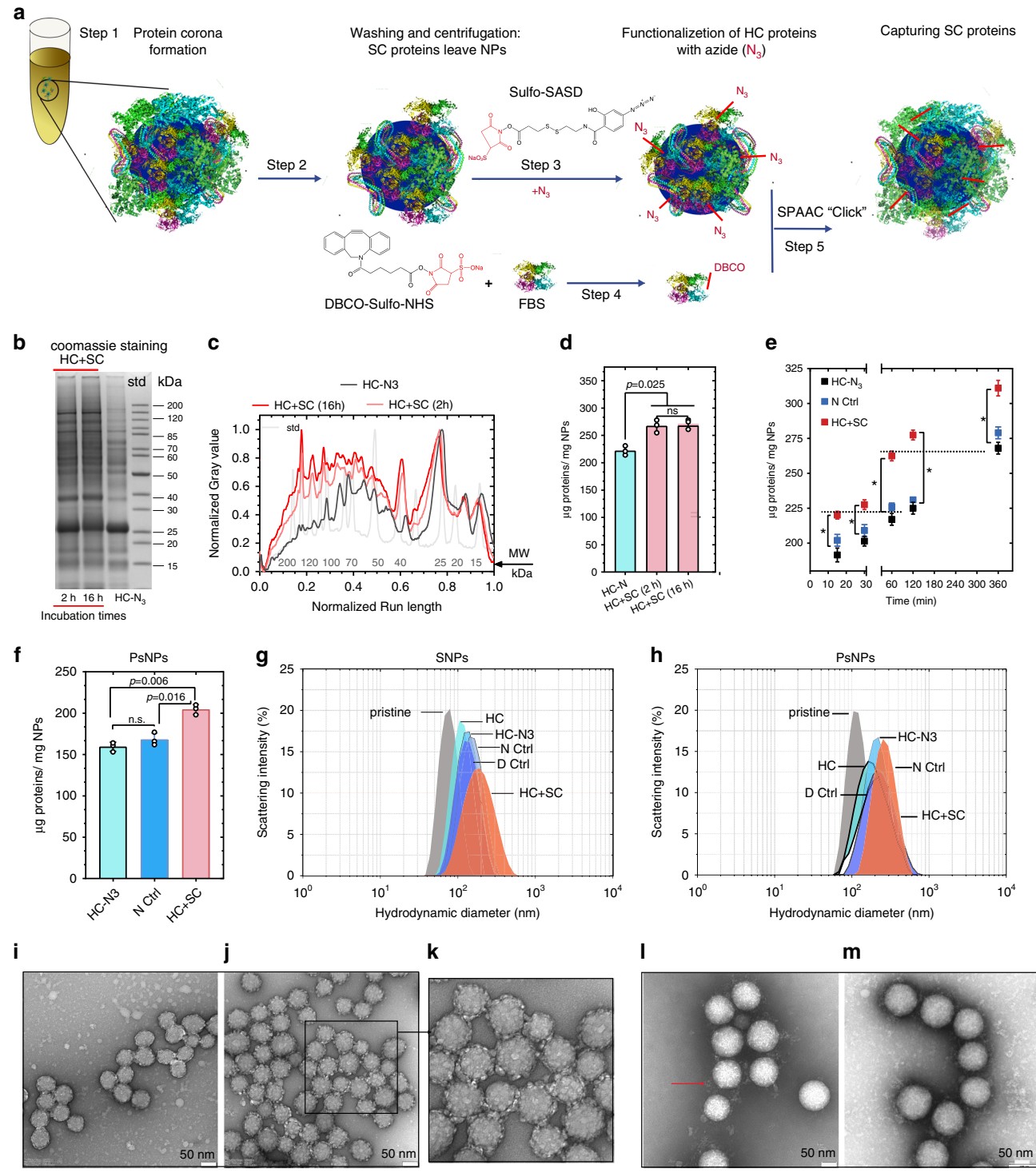

capture weakly interacting proteins. SDS-PAGE analysis revealed a change in protein patterns after the click reaction, which was further validated by capturing fluorescently labeled proteins (Supplementary Fig. 4). Quantification of total protein per nanoparticle further confirmed the increase in the mass of the corona proteins (~50 μg ml$^{-1}$ nanoparticles) after the click reaction. Extending the reaction time from 2 to 16 h did not result in a further increase in mass (Fig. 1d). The amount of SC proteins captured was positively correlated with the amount of HC proteins bound to the nanoparticles, which increased as a function of incubation time (Fig. 1e and Supplementary Fig. 5). Illustrating the applicability of this method to other types of

nanoparticles with different surface chemistry and charge, we observed comparable BCA and SDS-PAGE results for mixed charge amine-modified SNPs (SANPs, 75 nm), highly charged carboxyl-modified SNPs (SCNPs, 75 nm), and PsNPs (Fig. 1f and Supplementary Fig. 6).

Our click-chemistry approach to capturing SC proteins maintained a colloidally stable population of nanoparticle–corona complexes with a slightly increased hydrodynamic size and an increased particle heterogeneity (Fig. 1g–m, Table 1, and Supplementary Table 2). We further analyzed the formation of the protein corona by negative staining (TEM), which revealed a globular appearance of dehydrated proteins on SNPs (Fig. 1i–k

**Fig. 1 SPAAC click-chemistry reaction and characterization of nanoparticle–corona complexes. a** Schematic representation of capturing SC proteins. After protein corona formation (steps 1 and 2), the HC proteins were modified with $N_3$ by reacting with sulfo-SASD (step 3) followed by a SPAAC "click" reaction (step 5) with FBS-D proteins (prepared in step 4). **b–d** Effect of exposure time periods (2 and 16 h) in the click reaction evaluated by coomassie staining images (**b**) and densitometry analysis of SDS-PAGE gel (**c**), and quantification (**d**) of protein corona recovered from SNPs. The SDS-PAGE analysis was repeated three times independently with similar results. **e** Quantification of HC+SC proteins captured by click reaction on HC proteins formed on SNPs over different incubation times (15 min, 30 min, 1 h, 2 h, and 6 h). The SDS-PAGE image and densitometry analysis of the proteins are shown in Supplementary Fig. 5. **f** Quantification of HC+SC proteins captured by click reaction on PsNPs. Quantification data in **d–f** represented as the mean ± s.d. of three independent experiments ($n = 3$). For the multiple comparison, $P$ value was calculated by one-way ANOVA with Tukey post hoc test without any adjustment. *$P < 0.05$; n.s., not significant ($P > 0.05$). **g, h** Hydrodynamic analysis of nanoparticle–corona complexes, SNPs (**g**), and PsNPs (**h**). **i–m** Transmission electron microscopy (TEM) analysis of the SNPs@HC (**i**), SNPs@HC+SC (**j, k**), PsNPs@HC (**l**), and PsNPs@HC+SC (**m**). TEM analysis was performed three times independently with similar results. Scale bar, 50 nm. FBS-D: FBS proteins modified with DBCO, pristine silica nanoparticles (SNPs), pristine polystyrene nanoparticles (PsNPs), hard corona (HC), hard corona modified with azide (HC-$N_3$), FBS-D added to HC (D Ctrl), FBS added to HC-$N_3$ ($N_3$ Ctrl), FBS-D added to HC-$N_3$ (HC+SC), HC-coated SNPs (SNPs@HC), and HC-coated PsNPs (PsNPs@HC). Source data are provided as a Source data file.

---

**Table 1 Characterization of nanoparticle–corona complexes in buffer.**

| | Nanoparticle–corona complexes | Zeta potential ± SD (mV) | Hydrodynamic diameter ± SD (nm) (PDI) |
|---|---|---|---|
| SNPs | Pristine | −22 ± 3.4 | 81 ± 5.1 (0.01) |
| | HC | −20 ± 4.1 | 106 ± 5.7 (0.06) |
| | HC-$N_3$ | −25 ± 2.4 | 128 ± 6.3 (0.12) |
| | D Ctrl | −23 ± 3.7 | 133 ± 7.5 (0.08) |
| | $N_3$ Ctrl | −26 ± 3.3 | 121 ± 10.2 (0.07) |
| | HC+SC | −29 ± 2.9 | 152 ± 12.4 (0.17) |
| PsNPs | Pristine | −27 ± 2.1 | 110 ± 11.2 (0.02) |
| | HC | −26.2.3 | 155 ± 8.3 (0.1) |
| | HC-$N_3$ | −25 ± 3.1 | 168 ± 14.2 (0.15) |
| | D Ctrl | −23 ± 2.8 | 179 ± 13.6 (0.19) |
| | $N_3$ Ctrl | −26 ± 4.3 | 175 ± 9.2 (0.2) |
| | HC+SC | −25 ± 4.2 | 191 ± 15.3 (0.18) |

The average size of nanoparticle–corona complexes was determined using DLS, and the zeta potential measurement data processing was done by using Smoluchowski model. Zeta potential measurement was done in 10 mM sodium phosphate buffer, pH 7.4, containing 10 mM NaCl. Data shown correspond to mean ± s.d. of three independent experiments ($n = 3$).

and Supplementary Fig. 7), while a more diffuse appearance was observed for PsNPs (Fig. 1l, m and Supplementary Fig. 8). Image analysis of the SNPs confirmed a broadened distribution of the maximum Feret particle diameters with an increase in the mean size from 72 nm (HC) to 87 nm (HC+SC) (Supplementary Fig. 7d).

**SC/HC show different binding states of the same proteins.** Using the click reaction to fix the weakly interacting proteins in place, we were able to isolate SC proteins along with HC proteins by centrifugation and subject them to proteomic quantification by tandem mass spectrometry (LC–MS/MS). It should be mentioned that to avoid the potential for changes in the overall protein interactions with nanoparticles, e.g., highly modified BSA[34], we labeled the HC proteins with $N_3$ after formation on NPs and with a relatively low level of labeling. Further centrifugation and manipulation steps applied in this method, such as $N_3$ modification, did not significantly desorb HC proteins from nanoparticles; however, we believe that the slight effect of modifications on the SC profile is unavoidable.

We first calculated the copy number of each identified protein per nanoparticle following quantitation of the total protein mass (by BCA assays), nanoparticle mass (by fluorimetry), and emPAI-based relative mass percentages of proteins identified in LC–MS/MS. This allows a comparison of different samples without bias for large-sized proteins or the total protein input. Next, a cluster

of proteins specifically enriched in the HC+SC sample (based on the copy number of corona proteins per nanoparticle) was identified using bottom-up cluster analysis to construct two-way dendrograms along with a heatmap (Fig. 2a). In this approach, having a higher copy number than in the four control HC samples is not automatically considered to be indicative of an SC protein because a higher copy number may also be acquired by random variation. Therefore, the SC protein cluster is restricted to proteins that had a consistently lower (or zero) copy number in all of the four control samples without a large variation among them. The column dendrogram clearly separates the HC+SC sample from the rest. The row dendrogram reveals a putative SC cluster (colored in orange) characterized by specific enrichment of the proteins in HC+SC. For SNPs, 20 proteins were considered as SC proteins among the total of 80 proteins identified by LC–MS/MS, and only 4 out of the 20 SC proteins were uniquely captured after the click reaction (i.e., undetected in all HC controls), while the others were found in the HC controls to some extent (Fig. 2b). The total copy number of all proteins per NP increased 1.15-fold after the click reaction (cf. ~1.2-fold increase in the total protein mass in Fig. 1e), and the increase was mainly due to higher copy numbers of proteins belonging to the SC cluster (Fig. 2c), indicating the specific enrichment of these SC proteins. This can also be explained by the fact that the top 5 abundant proteins, which account for >50% of the total, remained the same even after the click reaction, whereas SC cluster proteins were ranked higher than before (e.g., 5.7-fold increase in the abundance of APOH, Table 2, "SNPs" and Fig. 2b). Importantly, the absence of highly abundant serum proteins, such as albumin in the SC cluster, shows that our click-chemistry method used in a competitive situation captures only proteins that are resident at the surface through a weak interaction with HC proteins and/or NP's surface and not proteins directly from the bulk.

Most of the SC proteins were also found in the HC, leading us to rethink our initial hypothesis that the SC is formed from different proteins from those in the HC. Our results rather indicate that the same proteins could have different binding constants, and that SC proteins are those capable of both stable and transient interactions. For simplicity, we describe the SC proteins as generally having two binding states: "hard" and "soft". The two binding states are assumed from the default presence of SC proteins in HC (hard binding) and upon "click" capturing of additional SC proteins in HC+SC (soft binding). Accordingly, we classified SC proteins into three types based on the relative copy numbers in the hard versus soft binding state: Type-1 SC proteins have more copies undergoing hard interactions, Type-2 SC proteins have similar copy numbers in the HC and SC, and Type-3 SC proteins have more copies undergoing soft interactions. This classification is visualized in Fig. 2d, e, where only the SC

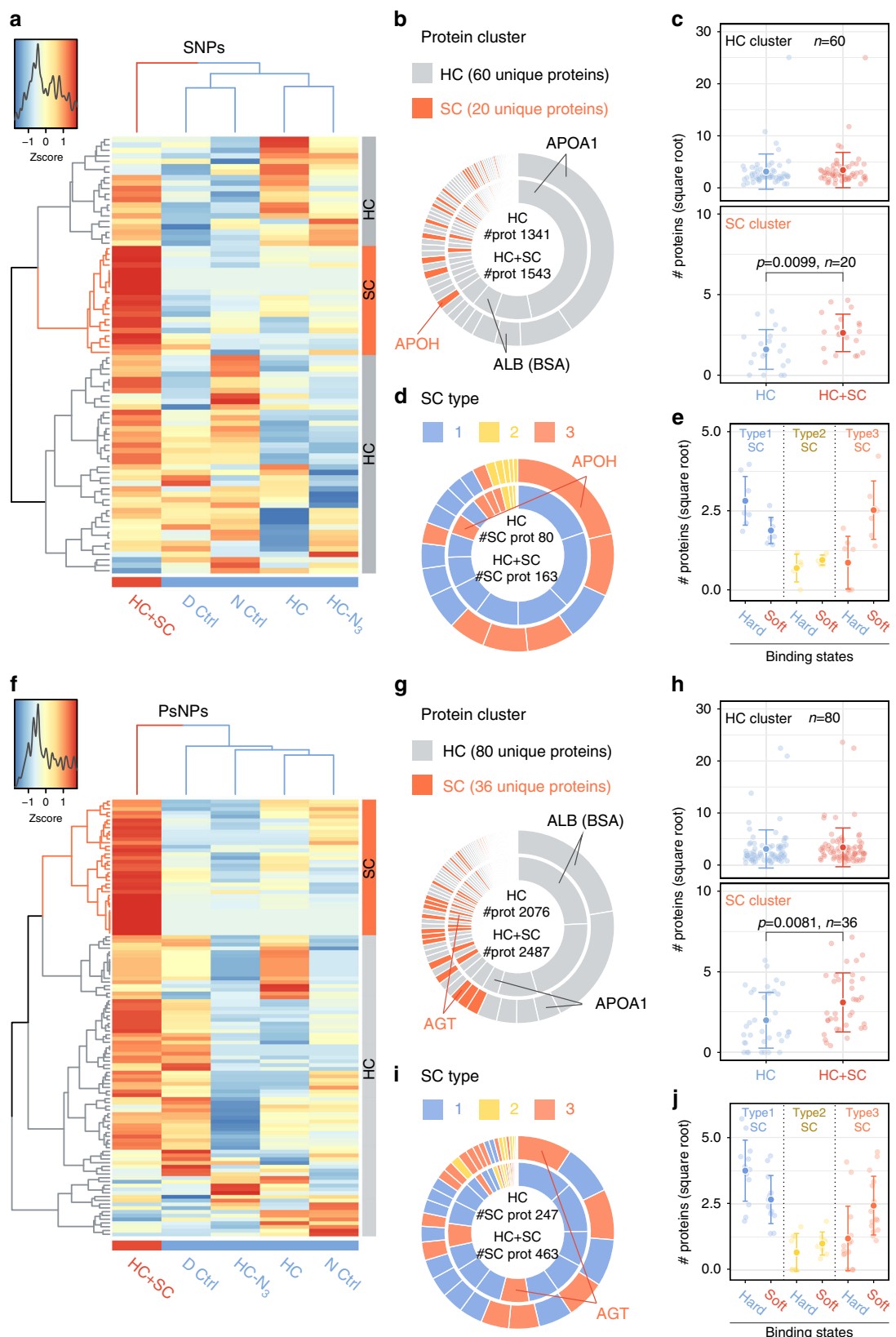

cluster proteins are displayed but with a different color code for each SC type. The total copy number of the SC cluster proteins increased ~2-fold after the click reaction with Type-3 SC proteins representing the major SC fraction, which were effectively captured (Fig. 2d) and thus overall increased in the copy number (Fig. 2e). In HC samples, the relative contribution of Type-1 SC proteins (blue) is larger than Type-3 SC proteins (orange), and vice versa in HC+SC samples. Of particular note, we observed a tendency for Type-3 SC proteins to have a higher GRAVY score and instability index, suggesting that these proteins are inherently less hydrophilic and less stable in serum (Fig. 3a). Neither the isoelectric point (Fig. 3a) nor multiparametric combinations of the

**Fig. 2 Identification of soft corona proteins and their classification according to the relative abundance in two binding states.** Shown are the results for SNPs (**a–e**) and PsNPs (**f–j**). **a, f** The relative abundance (z score) of each corona protein between samples is represented as a heatmap along with two-way unsupervised hierarchical clustering analysis. A color key along with the z-score distribution is depicted to the top left. See Supplementary Figs. 9 and 10 for the identity of corona proteins representing each row of the heatmaps. **b, g** The relative contribution of HC and SC cluster proteins to the total sum of copy numbers of corona proteins (#prot) in HC (averaged from all the four control samples) and HC + SC. The two doughnut charts represent the number percentages of each protein in HC (inner) and HC + SC (outer). Proteins of particular interest are annotated. **c, h** The copy number of proteins per nanoparticle is plotted for each unique protein from HC and SC clusters. **d, e, i, j** SC proteins are futher classified into three types, and their number statistics visualized in the same way as for HC versus SC (**b, c, g, h**). **e, j** The copy number of proteins per nanoparticle is plotted for the two binding states (soft and hard) of each SC protein characterizing the three different SC types. Values are shown for individual proteins (light colored) and as the mean ± s.d. (dark colored), where n is the number of unique proteins and labeled within each plot (**c, e, h, j**). Statistical significance was tested by two-sided Student's *t* test (**c, h**). Source data are provided as a Source data file.

**Table 2 Twenty most abundant proteins in HC and HC + SC samples and 20 most abundant SC proteins on SNPs and PsNPs.**

| No. | Top 20 proteins in HC samples | Top 20 proteins in HC + SC samples | Top 20 proteins in SC | Fold increase |
|---|---|---|---|---|
| | *SNPs* | | | |
| 1 | APO-AI/46% | APO-AI/40.5% | APOH | 5.71 |
| 2 | Hemoglobin fetal subunit beta/8.7% | Hemoglobin fetal subunit beta/8.9% | Complement C4 | 6.22 |
| 3 | Serum albumin/4.3% | Serum albumin/4.3% | UP2 | New |
| 4 | Hemoglobin subunit alpha/4% | Hemoglobin subunit alpha/4% | UP3 | New |
| 5 | Alpha-1-antiproteinase | Alpha-1-antiproteinase | UP1 | 3.31 |
| 6 | Kininogen-1 | Tetranectin | Cystatin-C | New |
| 7 | Plasminogen | Alpha-2-HS-glycoprotein | Fibulin-1 | 2.80 |
| 8 | Histidine-rich glycoprotein | Plasminogen | Collagen alpha-2(I) chain | 4.09 |
| 9 | Alpha-2-HS-glycoprotein | *APOH* | SERPINA11 protein | New |
| 10 | Apolipoprotein E | Serotransferrin | Fibronectin | 2.36 |
| 11 | Antithrombin-III | Histidine-rich glycoprotein | Alpha-S1-casein | 1.96 |
| 12 | Alpha-fetoprotein | Kininogen-1 | Thrombospondin-1 | 1.81 |
| 13 | Serotransferrin | *Antithrombin-III* | Complement C3 | 1.87 |
| 14 | Alpha-2-macroglobulin | Apolipoprotein E | Inter-alpha-trypsin inhibitor H3 | 1.60 |
| 15 | Protein S100-A12 | *Alpha-2-macroglobulin* | Inter-alpha-trypsin inhibitor H2 | 1.71 |
| 16 | Kininogen-2 | Alpha-fetoprotein | Hemopexin | 1.61 |
| 17 | Hemoglobin subunit beta | Insulin-like growth factor-binding 2 | Pigment epithelium-derived factor | 1.48 |
| 18 | Actin, cytoplasmic 1 | Apolipoprotein M | Prothrombin | 1.22 |
| 19 | Prothrombin | *Complement C3* | Antithrombin-III | 1.32 |
| 20 | Apolipoprotein M | Hemoglobin subunit beta | Alpha-2-macroglobulin | 1.25 |
| | *PsNPs* | | | |
| 1 | Serum albumin/24.2% | Serum albumin/22.3% | Angiotensinogen | 2.19 |
| 2 | Hemoglobin fetal subunit beta/20.3% | Hemoglobin fetal subunit beta/20.3% | Alpha-1-antiproteinase | 1.57 |
| 3 | Hemoglobin subunit alpha/3.8% | APO-AI/3.8% | Serpin family G member 1 | 2.35 |
| 4 | APO-AI/9.2% | Hemoglobin subunit alpha/3.7% | Serotransferrin | 1.59 |
| 5 | Plasminogen | Plasminogen | Tetranectin | 9.47 |
| 6 | Alpha-fetoprotein | Alpha-fetoprotein | Plasma serine protease inhibitor | 1.61 |
| 7 | Alpha-2-HS-glycoprotein | *Alpha-1-antiproteinase* | Cystatin -C | 22.22 |
| 8 | Alpha-1-antiproteinase | *Serotransferrin* | Complement factor I | 23.39 |
| 9 | Kininogen-1 | *Angiotensinogen* | Alpha-2-macroglobulin | 1.59 |
| 10 | Serotransferrin | Kininogen-1 | Complement factor H | 1.67 |
| 11 | Apolipoprotein E | Antithrombin-III | Insulin-like growth factor-binding 2 | 1.55 |
| 12 | Beta-2-glycoprotein 1 | *Plasma serine protease inhibitor* | Plasma kallikrein | 1.51 |
| 13 | Alpha-S1-casein | APOH | Secreted phosphoprotein 24 | 2.66 |
| 14 | Plasma serine protease inhibitor | *Serpin family G member 1* | Fetuin-B | 1.33 |
| 15 | Kininogen-2 | Alpha-2-HS-glycoprotein | Hemopexin | 4.93 |
| 16 | Antithrombin-III | Sulfhydryl oxidase | Coagulation factor XII | 1.81 |
| 17 | Fetuin-B | Apolipoprotein E | Kininogen-2 | 1.25 |
| 18 | Actin, cytoplasmic 1 | *Fetuin-B* | Protein AMBP | 1.28 |
| 19 | Angiotensinogen | *Kininogen-2* | Ig-like domain-containing protein | 11.98 |
| 20 | Beta-lactoglobulin | *Alpha-2-macroglobulin* | Fibulin-1 | 5.34 |

Entries in italics are the proteins among the 20 most abundant proteins that appear to be in both HC- and SC-binding states.

four protein parameters revealed particular trends for the Type-3 SC proteins (Supplementary Fig. 11). The number-weighted averages of the overall protein characteristics of the HC controls and the HC+SC sample did not show a particular propensity, indicating that the click reaction itself does not preferentially capture proteins with a specific parameter (Fig. 3c).

To look for a possible particle-type dependency, we applied the same method to analyze SC proteins on PsNPs. This revealed a slightly higher SC protein mass (1.2-fold increase after the click reaction) and more individual SC cluster proteins (36) than for SNPs (Fig. 2f–h). Notably, while there was a 30% overlap of HC cluster proteins between SNPs and PsNPs, only 7% of SC cluster

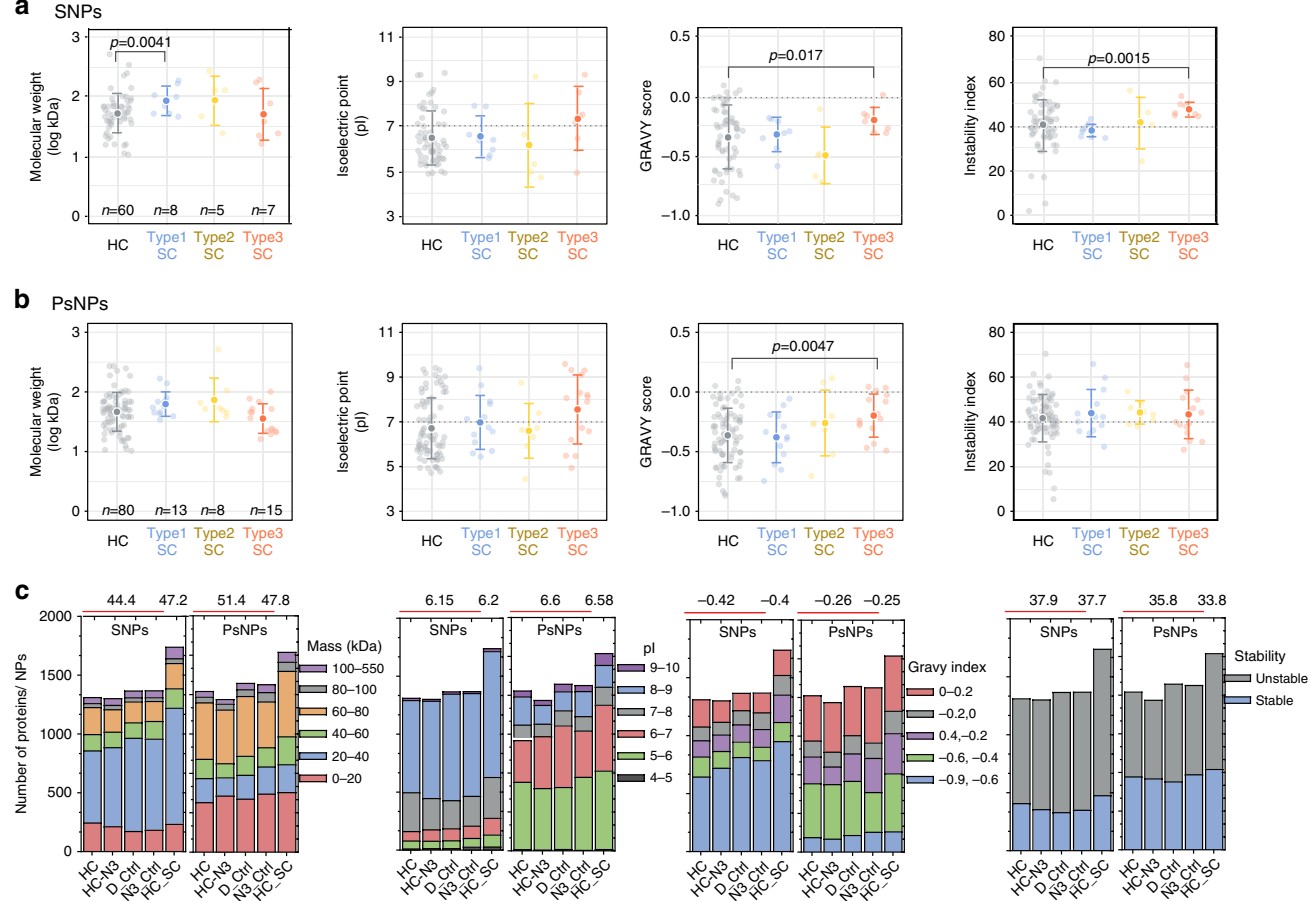

**Fig. 3 Parameter analysis of corona proteins.** Shown are the results for SNPs (**a**, **c**) and PsNPs (**b**, **c**). **a**, **b** Four protein parameters (molecular weight, isoelectric point, GRAVY score, and instability index) are compared between the HC proteins and each type of SC proteins. Values are shown for individual proteins (light colored) and as the mean ± s.d. (dark colored), where $n$ is the number of unique proteins and labeled within each plot. Statistical significance was tested by two-sided Welch's unequal variance $t$ test. **c** The number-weighted average of the four parameters characterizing the overall protein property of the corona is compared. The number-weighted protein parameters written above each plot were calculated for HC (average of four control samples) and HC + SC. Source data are provided as a Source data file.

proteins were common to both nanoparticle types (Supplementary Fig. 12). Type-3 SC proteins from PsNPs did not share properties, except for the GRAVY score that tended to have a higher value (less hydrophilic) than HC cluster proteins, as also observed for the SNPs (Fig. 3b and c and Supplementary Fig. 11). It should be mentioned that despite almost the same negative charge on both SNPs and PsNPs, the classification of corona proteins on SNPs and PsNPs was different (Fig. 3c), as thoroughly discussed in the previous studies[13]. This shows that other than electrostatic interactions between proteins and NPs with opposite charges, there can be electrostatic interactions between the exposed part of a denatured protein with the same charge, and NPs, or other interactions such as hydrophobic interactions, as PsNPs are more hydrophobic.

Weber et al.[26] by using an asymmetric flow field-flow fractionation (A4f) separation technique identified SC proteins on surfactant/polystyrene NP, and human serum albumin (HSA) was considered to be associated as a SC protein; similarly to our work, they also showed that the SC proteins are mainly present also in the HC. Kari et al.[27] found enrichment of stealth-mediating proteins toward phagocytes that were enriched as SC proteins on liposomes immobilized on an SPR sensor.

Here, we chose FBS as a culturing condition, which is routinely used for in vitro cell association studies. We expect to capture different HC and SC proteins by using different protein sources as

it is shown before that even a slight variation in the composition can substantially change the profile of the HC on NPs. Ezzat et al.[35] showed that different protein sources, such as human plasma, human bronchoalveolar lavage fluid, and fetal serum proteins, not only affect the composition of protein corona on viruses, but also affect viral infectivity and immune cell activation. Hajipour et al.[36] also confirmed that protein pattern on silica and polystyrene nanoparticles differed both in terms of composition and amount with various diseases.

Based on the copy numbers of total proteins calculated in this study, we were also able to derive the surface coverage of total protein cross-section areas per nanoparticle. After the click reaction, the theoretical coverage ratio of NPs by corona proteins increased from 0.80–1.01 to 0.97–1.25 (for SNPs) and from 0.70–0.99 to 0.93–1.20 (for PsNPs), depending on the orientation of proteins on the NPs. It should be noted, however, that our click-chemistry approach relies on the presence of an azide group on a HC protein within the reach (~1.8 nm) of the DBCO moiety of SC proteins, thus likely underestimating the amount of SC proteins, particularly for low HC coverages. Therefore, it is likely that the complete SC has still not been captured here. Nevertheless, the derived coverage ratio is consistent with previous studies claiming that the corona formed from serum consists essentially of a monolayer[1,37]. On this basis, we hypothesize that both HC and SC proteins coexist within a

loosely defined monolayer covering the nanoparticle surface, rather than in separate layers, and that SC proteins may dissociate from the surface during centrifugation. This fits well with our two binding-state models introduced above, and suggests that the monolayer of proteins becomes less dense as SC proteins dissociate, partially exposing the bare surface of the nanoparticle.

**APOH shows multiple binding strengths at HC-coated SNPs.** Apolipoprotein H (APOH) is among the most enriched proteins in Type 3, representing the major SC proteins on SNPs but not on PsNPs (Fig. 2 and Table 2). To test the ability of APOH to bind SNPs in a "hard" and "soft" state, a series of competition studies were performed using BSA and FBS. BSA was selected as the direct competitor since it is a HC cluster protein and has a higher affinity for the surface of SNPs than APOH. We started with uncoated SNPs, and confirmed that APOH shows weaker binding to SNPs in the presence of the competitor BSA (Fig. 4a, b). We then let the HC preassemble on SNPs and captured APOH's soft interactions through the click-chemistry approach. Unlike the simple race for the bare surface, APOH was effectively enriched by the click reaction, outcompeting BSA with a negligible influence of the concentration ratio between the two (Fig. 4c, d). The low competitiveness of BSA for soft interactions with SNPs was also supported by the reduced capture of BSA via the click-chemistry reaction, despite its high abundance, in the presence of FBS (Fig. 4e, f). Interestingly, in a parallel competition study, the disease-related α-synuclein (α-Syn) showed weak but specific binding to the HC proteins on SNPs unaffected by the presence of FBS proteins (Supplementary Fig. 14). Using different eluting solutions (reducing and nonreducing sample buffers and DTT solution) confirmed that a reducing agent is necessary to elute the majority of α-Syn from the nanoparticles. This confirms that α-Syn is captured mainly through the click reaction, and DTT can reduce the disulfide bridge in the sulfo-SASD structure to release immobilized α-Syn.

To study the soft binding property of APOH without click reactions, we utilized an immobilised-nanoparticle SPR setup to quantify the interaction of label-free APOH with FBS proteins deposited onto SNPs[38] (Fig. 4g). In this setup, SNPs were first immobilized on repellent PLL-g-PEG, as a protein-resistant polymer, which reduces unspecific protein binding directly on the SPR biosensor. Then, protein corona was formed by injecting 1% FBS onto the immobilized NPs. When APOH was added to SNPs with protein corona in different concentrations (1, 5, 15, 50, and 150 μg ml$^{-1}$), a fraction of APOH remained associated with SNPs in a concentration-dependent manner (Fig. 4h). $K_d$ and $k_{off}$ values (Supplementary Table 3) were obtained by fitting a two-dimensional equation to the APOH data (Fig. 4i), which shows a major population (88 ± 4.4% of the signal) with high $K_d$ values and a fast off-rate and a small smeared population (9.5 ± 3.5% of the signal) with low $K_d$ values and a slow off-rate. This study confirms the differential binding strength of APOH, where the major and minor fractions have high and low dissociation rates, representing soft and hard states of interactions on SNPs, respectively. The fraction of APOH being removed at rinsing is also considered as another population with even lower binding strength. A rapid drop of up to 60% of the adsorbed APOH on FBS-coated SNPs was seen in 50 μg ml$^{-1}$ APOH, while this drop is only 6% for adsorption of FBS on SNPs, indicating that the FBS proteins are tightly adsorbed to the NPs (Supplementary Fig. 15).

**SC proteins modulate cell association of nanoparticles.** Conceptually, pristine nanoparticles spontaneously bind to cell membranes in a nonspecific manner in serum-free conditions, while protein coronas, in general, reduce this nonspecific

interaction by blocking the surface[23] and potentially increase specific interactions. For cell association (CA) via specific interactions, significant focus has been given to the HC proteins that have a residence time sufficient to support biological interactions (e.g., receptor ligation)[22,39,40]. To elucidate the contribution of SC on CA, seen as the net effect of nonspecific interactions of cells with the particle surface accessible by the dynamic SC proteins and specific interactions with the HC and SC (Supplementary Fig. 17), the click-chemistry method was employed, which hardens and transforms the SC proteins to HC, providing them a biologically relevant residence time. This is a modulation of the real condition for nanoparticle–cell interaction studies, but it provides a means for studying the effect of the dynamic exchange of soft corona proteins on cell associations. Phagocytic-differentiated macrophage-like THP-1 cells (THP-1 macrophages), which express a family of cell surface recognition receptors[41], and human brain cerebral microvascular endothelial cells (hCMEC/D3), as a nonphagocytic counterpart, were used in this study. It should be noted that we use bovine-origin serum with these human-derived but FBS-conditioned cell lines.

A preliminary study showed that cells readily interact nonspecifically with bare particle surfaces of pristine nanoparticles, which in general decreases from serum-free (SF) conditions to BSA media and then decreases further in FBS media, more significantly for SNPs (Supplementary Fig. 18 and Fig. 5a). The effect of FBS on CA for the two particle types (over time of HC formation) was markedly different (Fig. 5a–d), suggesting some involvement of specific interactions for SNPs. PsNPs, where BSA was the dominating protein in the corona, showed a systematic reduction of CA as the HC formed for both cell types, more significantly for late corona (2 and 6 h). In contrast, SNPs, where APO-A1 was the major corona protein, showed an increased (for THP-1 cells) or unchanged (with hCMEC/D3 cells) CA as the HC formed, suggesting a different role for the dominant HC proteins in driving cell adhesion. Interestingly for SNPs, an opposite trend is seen when FBS-formed HC is studied in BSA media (with a pattern more similar to PsNPs).

When particles are introduced to the protein-containing cell media, the short-lived SC protein corona may take shape either from the BSA or FBS, respectively, in the background source. The artificial hardening of the SC proteins (HC+SC), which turns off the dynamic exchange of SC proteins, generally decreased the CA of PsNPs in both cell types and media conditions (FBS or BSA) compared to particles with HC and short-lived dynamic SC, while SNPs showed decreased CA in THP-1 macrophages in FBS medium and hCMEC/D3 cells in the BSA medium (Fig. 5a–d). No difference in the localization of nanoparticles in the cells was observed by confocal microscopy (Fig. 6). The hardening was also confirmed by a reduced ability of proteins to leave NPs in a serum-free medium (Supplementary Fig. 19a, b), or for proteins to be exchanged with medium proteins such as albumin (Supplementary Fig. 19c), which may explain why nanoparticles with hardened HC were less affected by changing media than pristine nanoparticles. The artificial cross-linking gives a more significant effect on CA than hardening made through evolution from a loosely attached toward a largely irreversibly attached protein during incubation[42], as even long formation times for HC do not reduce the amount of SC proteins detected (Fig. 1e).

To confirm if specific proteins in HC and SC can enhance cell adhesion, BSA was cross-linked onto HC on both SNPs and PsNPs, and APOH onto HC on SNPs. Surface-bound BSA has been reported to bind scavenger receptors on macrophages[22], and BSA specifically binds to FcRn receptors in endothelial cells[43,44]. While BSA cross-linked SNPs increased the CA in THP-1 cells, there was no effect on BSA-linked PsNPs (Fig. 5e, f). The increase in the CA of SNPs with cross-linked BSA implies

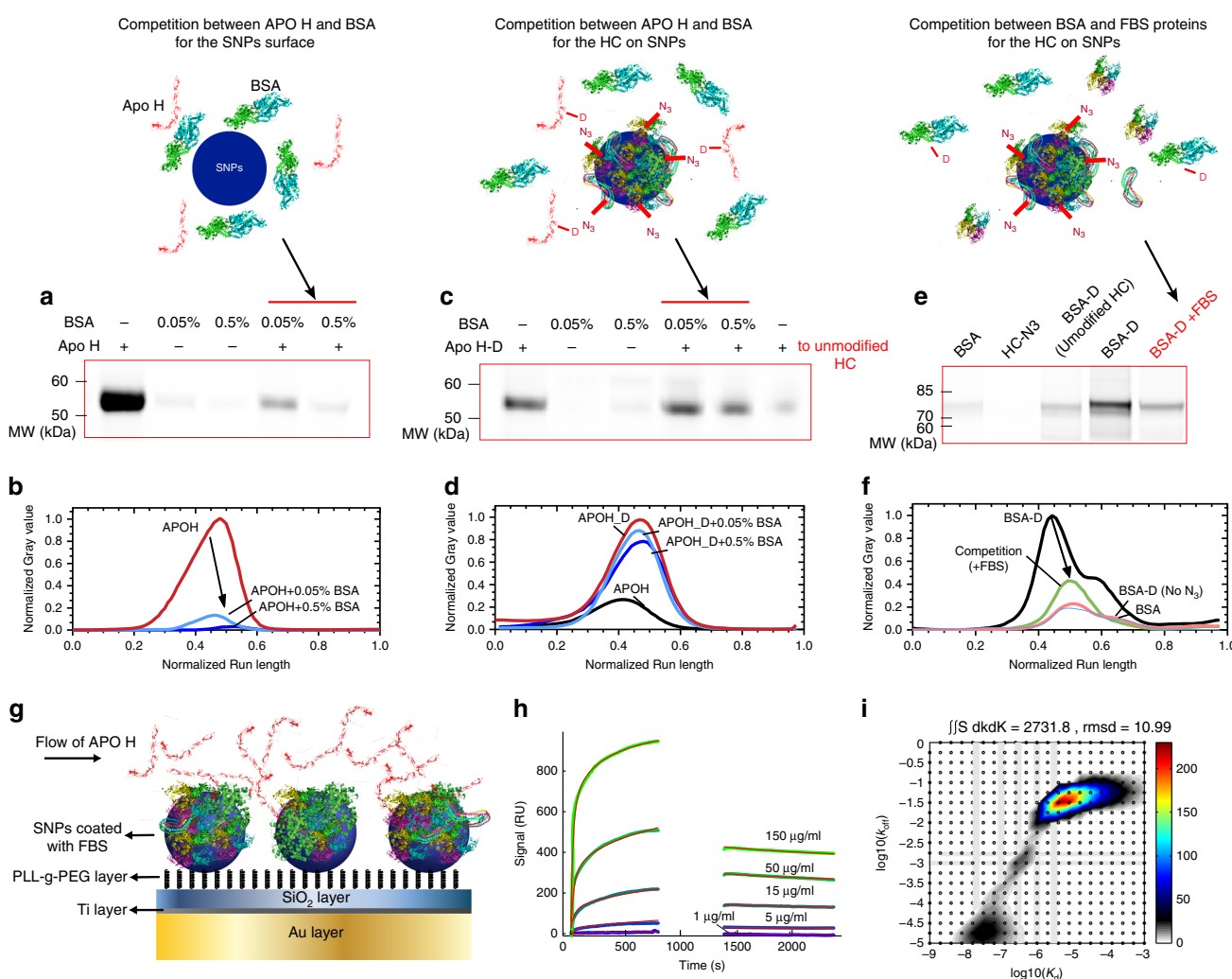

**Fig. 4 CY5-labeled APOH and BSA competition for the surface of SNPs and HC. a–d** Fluorescence images and densitometry analysis of cut-out SDS-PAGE gels of the addition of 30 μg ml⁻¹ of APOH-CY5 protein corona on SNPs (**a**, **b**) and 30 μg ml⁻¹ of APOH-CY5-DBCO on HC-$N_3$ on SNPs (**c**, **d**) formed in the presence and absence of either 0.05% or 0.5% BSA. APOH-CY5-DBSO was added to SNPs@HC without $N_3$ as a control sample in (**c**). **e**, **f** The competition study between BSA and other FBS proteins for SNPs@HC-$N_3$. Fluorescence image of a cut-out SDS-PAGE gel of proteins recovered from the corona formed on SNPs (**e**) and its densitometry analysis (**f**). In this experiment, BSA-CY5-DBCO alone or spiked with FBS proteins was added to SNPs@HC-$N_3$. The addition of BSA-CY5 to HC-$N_3$ and BSA-CY5-DBCO to HC without $N_3$ was done as controls. The complete SDS-PAGE gels are shown in Supplementary Fig. 13. **g–i** Surface plasmon resonance (SPR) characterization of APOH interaction with SNPS@HC. The schematic representation of the SPR characterization is shown in (**g**). First, SNPs were immobilized on a protein-resistant polymer, PLL-g-PEG, creating an array of SNPs on a protein-resistant background. Then, protein corona was formed by injecting 1% FBS onto the immobilized NPs (shown in Supplementary Fig. 15). Subsequently, APOH was injected in different concentrations (1, 5, 15, 50, and 150 μg ml⁻¹) (**h**). Two-dimensional fits were applied to the data to calculate $K_d$ and $K_{off}$ values for different populations of APOH binding to the SNPs@HC (**i**). For the fitting, data from around the rinsing were omitted, due to too few data points. The complete data of two technical repeats are shown in Supplementary Fig. 15. Source data are provided as a Source data file.

that surface-bound BSA is recognized by cell receptors, and since BSA is the main protein forming the protein corona of PsNPs, cross-linking BSA did not significantly change its already relatively high contribution. APOH as a major but weakly interacting protein in the SNP corona is described as an opsonin for the mononuclear phagocyte systems, which also binds phospholipids in membranes[45,46]. Cross-linked APOH could increase the CA of SNPs in both phagocytic and nonphagocytic cell lines (Fig. 5g, h); however, the addition of APOH to BSA medium decreased CA of pristine nanoparticles and did not affect CA of nanoparticles with HC.

CA is also related to the differences between cell types (e.g., phagocytic vs. nonphagocytic, cancer vs. normal cells, and monocytes vs. macrophages). THP-1 cell lines exhibit enhanced

expression of macrophage surface markers as compared to primary monocytes and macrophages[39]. In this study, we used PMA to induce differentiation of THP-1 cells to a macrophage phenotype to represent a cell type expressing a wider variety of receptors for phagocytosis as in primary macrophages, although some differences in the gene-expression profile exist between the two (e.g., a higher expression level of scavenger receptor A)[41]. Endothelial cells such as hCMEC/D3 cells, on the other hand, are active in endocytosis facilitating cross-cellular transport of nutrients and other biomolecules[47], notably including FcRn receptors for BSA[42].

To evaluate the level of aggregation of the NPs in the cell medium (RPMI+10% FBS), which can influence the cell association of NPs, the size distribution of nanoparticle–corona

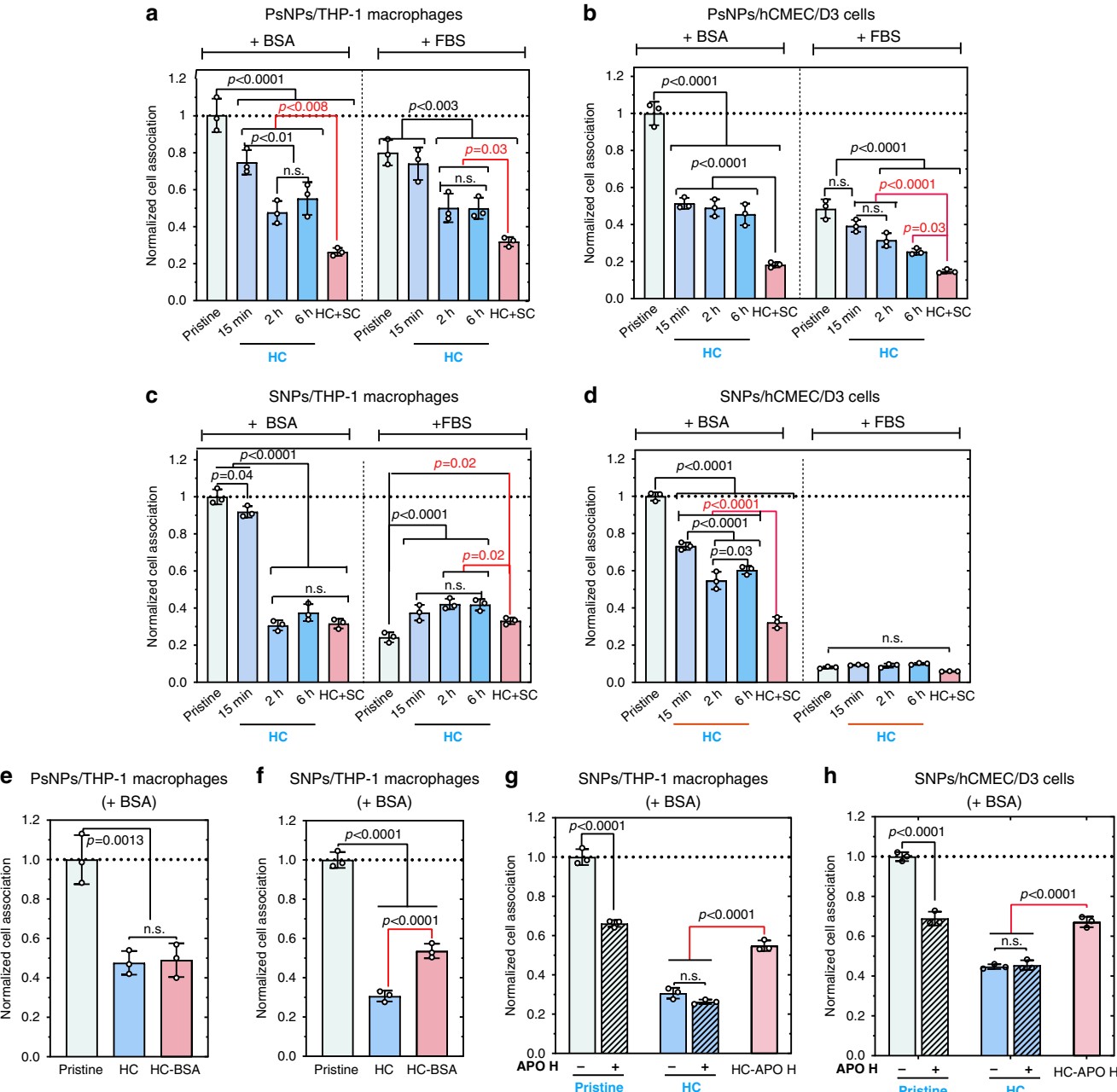

**Fig. 5 Cell association of nanoparticle–corona complexes. a–d** Flow cytometry was used to quantify the cell association of 50 µg ml$^{-1}$ of nanoparticle–corona complexes in THP-1 macrophages (PsNPs, **a** and SNPs, **c**) and hCMEC/D3 cells (PsNPs, **b** and SNPs, **d**). The cells were exposed to the pristine NPs, coated with HC formed over different FBS exposure times (15 min, 2 h, and 6 h), and with HC + SC, for 4 h in RPMI containing BSA or FBS. **e**, **f** THP-1 macrophages were exposed to PsNPs, PSNPs@HC, and PsNPs@HC-BSA (BSA is cross-linked on HC by using a click reaction) (**e**) and SNPs, SNPs@HC, and SNPs@HC-BSA (**f**) for 4 h in RPMI. **g**, **h** THP-1 macrophage cells (**g**) and hCMEC/D3 cells (**h**) were exposed to SNPs or SNPs@HC for 4 h in RPMI containing BSA with or without 30 µg ml$^{-1}$ APOH. The cells were also exposed to SNPs@HC_APOH (APOH is cross-linked on HC by using a click reaction). The flow cytometry data were normalized to the pristine nanoparticle values in the RPMI supplemented with BSA. Bars show mean ± s.d. of three biologically independent experiments. For the multiple comparison in (**a–d**), $P$ value was calculated by two-way ANOVA with Tukey post hoc test without any adjustment. $P$ value in (**e–h**) was calculated by one-way ANOVA with Tukey post hoc test. n.s., not significant ($P > 0.05$). The cell gating data, which were used to identify single cells, are shown in Supplementary Fig. 16. Source data are provided as a Source data file.

complexes was determined using DLS (Supplementary Table 4). The results showed that the particles in the cell medium were still stable against aggregation. We expect that the accumulation of particles observed in the confocal images occurred after CA.

We interpret the reduced CA by cross-linking of SC from FBS as indicating that the dynamic SC proteins keep regions of the particle surface free from HC and available for interaction with cells, despite the formed protein corona, allowing nonspecific

binding of the nanoparticle surfaces to cell membranes, dependent on their surface chemistry. Cross-linking would then decrease both the possibility for corona proteins to be exchanged with medium proteins and to reveal empty patches on the nanoparticle surface and enable nonspecific cellular association. Here we show that SC proteins can contribute nonspecifically to CA by revealing bare NP surfaces, while hardening of individual SC proteins can contribute to CA through specific interactions.

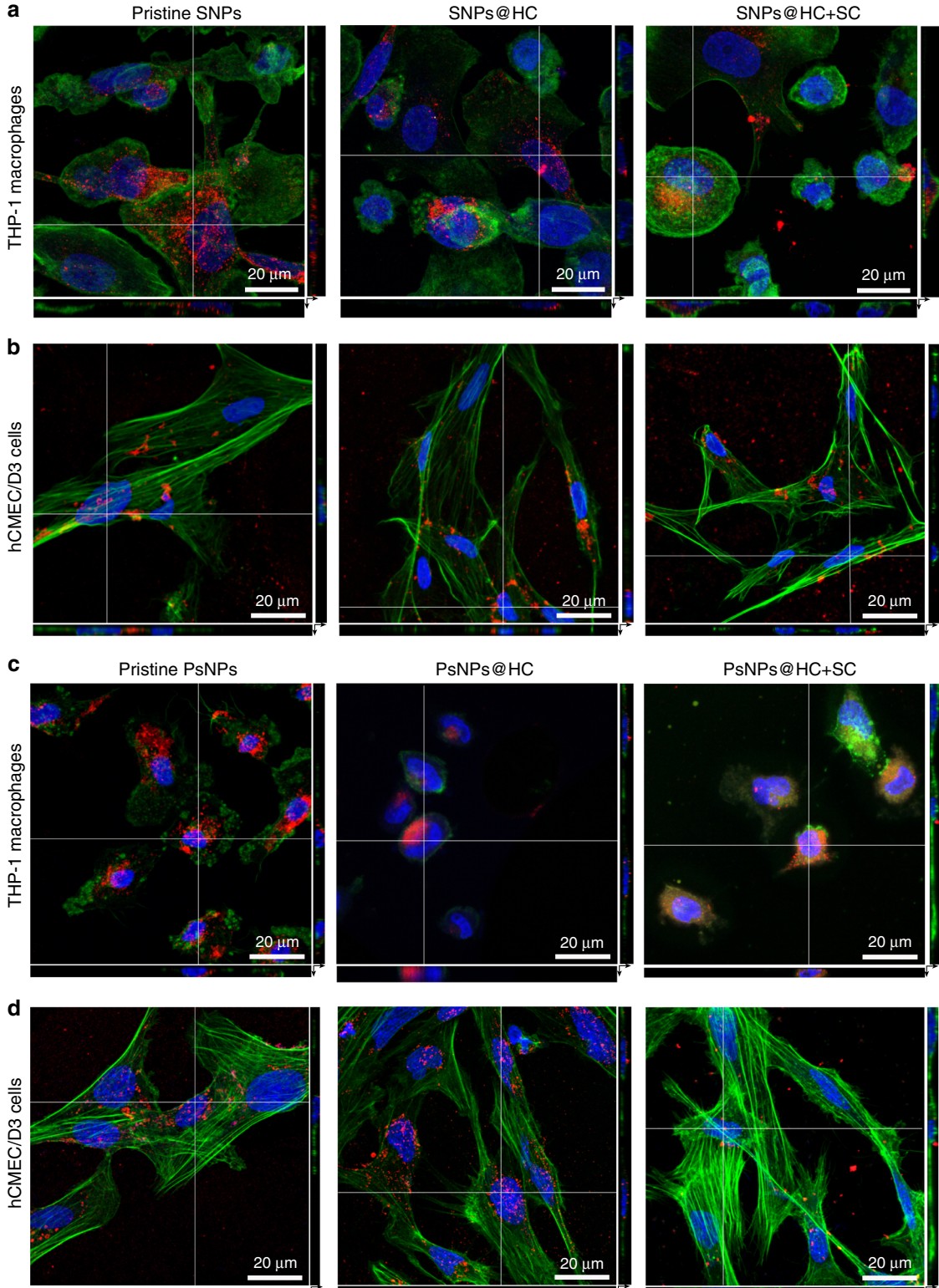

**Fig. 6 CLSM images confirm the uptake of nanoparticles. a**, **b** Orthogonal views of 3D stacks of CLSM images of SNP–corona complexes confirm the uptake of nanoparticles in THP-1 macrophages (**a**) and hCMEC/D3 cells (**b**) in RPMI containing BSA. **c**, **d** Orthogonal views of 3D stacks of CLSM images of PsNP–corona complexes confirm the uptake of nanoparticles in THP-1 macrophages (**c**) and hCMEC/D3 cells (**d**) in RPMI containing BSA. The images demonstrate that by vigorous washing of nanoparticles, all noninternalized particles had been removed from the cell surface prior to flow cytometry analysis, and they are internalized by both cell types. In CLSM images, no differences in the localization of NPs inside the cells were observed. The cells grown on collagen-coated coverslips were fixed, and then the cell nuclei (blue) and the actin filaments (green color) were stained with Hoechst and phalloidin, respectively. The FITC -labeled nanoparticles are shown in red color. Scale bar, 20 μm. The CLSM analysis was repeated three times independently with similar results. Source data are provided as a Source data file.

While Weber et al.[26] showed that only the HC proteins on surfactant/polystyrene NPs directly influenced the uptake of NPs by HeLa cells and SC did not alter the biological behavior, Kokkinopoulou et al. showed that SC reduces the cellular uptake of nanoparticles in comparison to HC-coated nanoparticles[24]. Kari et al. also suggested that SC of loosely interacting proteins on liposomes contributes to the stealth properties as a component of the biological identity[27].

## Discussion

We describe here a general and simple capture process based on click chemistry, which enabled the identification of weakly interacting proteins along with the long-lived protein corona forming around nanoparticles in complex media. For four particle types with different physicochemical properties, we find that the majority of the captured proteins are not unique to SC, but also present in the HC, indicating that the same proteins can have both strong and weak interactions with nanoparticles or pre-adsorbed neighboring proteins. As demonstrated using four types of model nanoparticles (SNPs, SANPs, SCNPs, and PsNPs), this method is general and can be applied to different types of nanoparticles without bias for the enrichment of specific types of proteins. Centrifugation is not a prerequisite for this method, and other separation methods, such as magnetic separation or size-based separation techniques, can be applied on different nano-particles. Our method is applicable to nanoparticles whose HC profiling is possible by using some kind of separation method.

SC proteins can thus be classified into three types based on the relative distribution between the hard and soft binding states (Fig. 7a). Given that the same proteins are interacting with the same surface and thus surface property but with a differential binding strength, we propose that neighboring proteins can restrict the hardening of SC proteins sterically limiting the surface area available for protein unfolding. Notably, the same SC protein may be categorized differently depending on the nanoparticle properties (e.g., APOH is Type-3 SC for SNPs, but HC for PsNPs), and therefore soft interactions cannot be generalized but may rather be surface-specific. By artificially hardening the SC proteins, we were able to turn off their dynamic nature of dissociation revealing a role in cellular interactions as "caretaker" proteins that can both prevent irreversible blocking of the surface and also allow higher-affinity interactions between cell membranes and transiently bare nanoparticle surfaces (Fig. 7b). The potential for cell–NP surface interactions suggests that the properties of the bare particle surface can still directly influence CA even after a fully developed protein corona is formed.

## Methods

**Nanoparticles**. The 70-nm fluorescent plain amine and carboxyl-modified silica nanoparticles (sicastar-greenF, fluorescein isothikcyanate-labeled; ex/em = 485/510 nm) were purchased from micromod Partikeltechnologie GmbH (Germany). The fluorescent carboxylate-modified polystyrene nanoparticles (FluoSpheres™ Carboxylate-Modified Microspheres, 0.1 μm, yellow-green fluorescent (505/515)) were purchased from Thermo Fisher.

**Click-chemistry reagents**. Sulfo-SASD (sulfosuccinimidyl-2-(p-azidosalicyla-mido) ethyl-1,3-dithiopropionate) was purchased from G-biosciences. Dibenzo-cyclooctyne (DBCO)-Supho-NHS and DBCO-Sulpho-Cyanine5 were purchased form click-chemistry tools and Genabioscience companies, respectively. Poly (L-lysine)-graft[3.5]-Poly(ethylene glycol)(2) (PLL-g-PEG) was purchased from SuSoS Ag.

**Fluorescent reagents**. Sulpho-NHS-Cyanine5 was purchased form Lumiprobe. Hoechst 33342 Solution (20 mM) was purchased from Thermo Fisher. Phalloidin–Tetramethylrhodamine B isothiocyanate was purchased from Sigma-Aldrich.

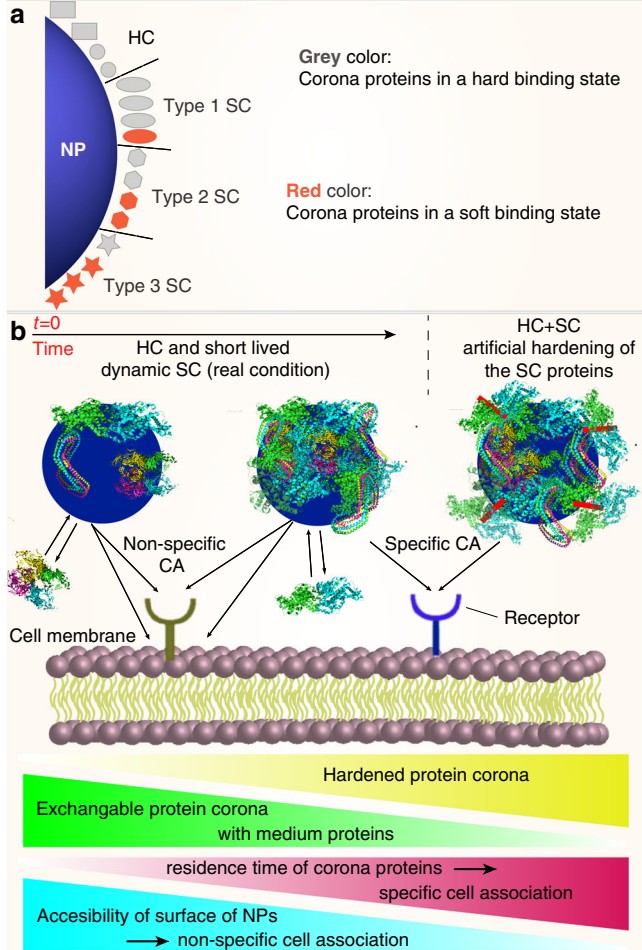

**Fig. 7 Schematic representation of the proposed model for the formation of hard and soft coronae. a** Three types of SC proteins were identified by LC–MS/MS analysis after the click reaction. The SC proteins are mainly the same as HC proteins, while the process of hardening does not define the protein corona composition, but rather changes the binding strength of the already-formed SC. **b** Hardening of HC through either more incubation time or cross-linking SC by using a click reaction prepares a biologically relevant residence time for corona proteins, and reduces the exchange of corona proteins with medium proteins and accessibility of the surface of NPs. The net effect of specific and nonspecific associations will determine the fate of nanoparticles in contact with cells. CA cell association.

**Cell media**. Penicillin–Streptomycin, Gibco, heat-inactivated Fetal Bovine Serum (FBS), and RPMI 1640 Media were purchased from Thermo Fisher. PMA Phorbol 12-myristate 13-acetate was purchased from Sigma-Aldrich.

**Nanoparticle incubation with FBS**. FBS was first centrifuged at $16,000 \times g$ for 3 min to remove any insoluble aggregates. Then, the protein supernatant and NPs solutions were preincubated at 37 °C for 10 min before mixing. The NPs were then exposed to FBS for different time points (15 min, 30 min, 1 h, 2 h, and 6 h) in darkness at 37 °C in Protein Lobind tubes. The ratio of total particle-surface area to FBS volume was kept constant for all nanoparticles. About 0.4 mg of SNPs (70 nm) and 0.3 mg of PsNPs (100 nm) were used. The nanoparticle–corona complexes were isolated from unbound and loosely bound FBS proteins by centrifugation at $18,000 \times g$, 20 min. Pellets were washed three times with PBS by centrifugation at $20,000 \times g$, 20 min, and resuspended in PBS for further analysis.

**Azide modification of HC proteins**. To generate azide-functionalized HC proteins, sulfo-SASD at different concentrations (0, 0.018, 0.036, 0.09, 0.18, 0.35, 0.55, and 1.8 mM) was added to the nanoparticles@HC at a final concentration of 0.4 mg ml$^{-1}$. Sulfo-SASD contains a dithiol in the structure that can be cleaved by a reducing agent for electroporation analysis. The complex was incubated at 37 °C

for 1 h to allow the reaction between the sulfo-NHS and amine groups on HC proteins. The azide-modified nanoparticle–corona complexes were separated from unreacted sulfo-SASD by centrifugation at 20,000 × g, 20 min. The pellet was washed three times with PBS and then resuspended in PBS for further steps. To characterize the azide groups on HC proteins, a click reaction between the azide groups and DBCO-sulfo-Cy5 was employed. By measuring the excited fluorescence (at 646 nm) of proteins at 664 nm, the minimum number of azide groups was calculated on HC proteins, assuming that each DBCO-sulfo-CY5 reacts with one sulfo-SASD. The number of HC corona proteins was measured by two methods: (1) by calculating the average molecular weight of proteins on SDS-PAGE and quantification of proteins by BCA assay and (2) LC–MS/MS analysis. In all steps, the number of nanoparticles was measured by reading the fluorescence of NPs.

For nanoparticles that interfere with the analysis, the proteins should be first eluted from nanoparticles, and then the click reaction between DBCO-sulfo-Cy5 can be performed (Supplementary Fig. 1a). We performed the eluting protocol for SNPs and compared the results with a noneluting protocol. To elute the proteins, the nanoparticles were resuspended in 500 μl of 1% acetic acid and incubated overnight. Then, the solution was removed by a centrifugal evaporator, and the proteins in the pellet were quantified by BCA assay (which is explained in the following). Then, the proteins were incubated with DBCO-sulfo-CY5 for the click reaction, and the unreacted dyes were eliminated by using a Sephadex G-25 in PD-10 Desalting Columns (GE Healthcare Life Sciences). The results of the BCA assay and click reaction showed the same efficiency for both eluting and noneluting protocols.

**Modification of proteins**. The reactive DBCO sulfo-NHS at various final concentrations (0, 0.2, 0.4, and 0.8 mM) was added to a tube containing 10% FBS solution (4.2 mg ml⁻¹) at PBS (pH 7.4). The molar ratio of the cross-linkers to proteins (if we assume all proteins are BSA) at the selected concentrations is equal to 0, 2.5, 5, and 10. The system was then incubated at 37 °C for 1 h to allow the reaction between the sulfo-NHS and amine groups on proteins. The reaction was quenched by Tris at a final concentration of 50 mM. The unreacted DBCO molecules were eliminated by a PD-10 Desalting Column. To make fluorescently labeled DBCO-modified FBS proteins or individual proteins (BSA or APOH), proteins were incubated with both DBCO-sulfo-NHS and sulfo-NHS-Cy5 (at the molar ratio of cross-linkers to proteins of 5). The degree of labeling (DOL) of proteins was calculated by measuring the UV spectrum of conjugates using the following equation:

$$\mathrm{DOL} = \frac{A_{\max} \times \varepsilon_{280}}{(A_{280} - A_{\max} \times \mathrm{CF}) \times \varepsilon_{\max}} \quad (1)$$

where $A_{\max}$ and $A_{280}$ are absorbances of the conjugate solution measured at 280 nm and at $\lambda_{\max}$ of the cross-linker or dye, respectively. $\lambda_{\max}$ values for DBCO sulfo-NHS and NHS-sulfo-CY5 are 309 and 646 nm, respectively. $\varepsilon_{280}$ and $\varepsilon_{\max}$ are the extinction coefficient of proteins (for FBS proteins, we used $\varepsilon_{280}$ value of albumin, 66,433 M⁻¹ cm⁻¹) and DBCO sulfo-NHS or NHS-sulfo-CY5, whose values were taken as 12,000 and 271,000 M⁻¹ cm⁻¹, respectively. CF is the correction factor of each cross-linker or dye, which is required to eliminate the contribution of the dye at 280 nm. $A_{\max}$ and $A_{280}$ are absorbances of the conjugate solution measured at 280 nm at $\lambda_{\max}$ of the cross-linker or dye.

**Cross-linking SC proteins on HC proteins**. In order to capture weakly interacting proteins on nanoparticle–corona complexes, the azide-modified particles were incubated with DBCO-modified FBS (FBS-D) for 2 h at 37 °C. The azide-modified nanoparticle–corona complexes were separated from free proteins by centrifugation at 20,000 × g, 20 min. The pellet was washed three times with PBS and then resuspended in PBS for further steps.

**Characterization of nanoparticle–corona complexes**. Nanoparticles were characterized by transmission electron microscopy (TEM), dynamic light scattering (DLS), and zeta potential measurements. Both zeta potential and hydrodynamic diameter of nanoparticles were measured by a Malvern Zetasizer Nano (Malvern Instrument Ltd., UK) with a laser wavelength of 633 nm in 10 mM sodium phosphate buffer at pH 7.4. For the calculation of zeta potential, data processing was done by using Smoluchowski model[48]. For TEM analysis, nanoparticles were loaded onto glow-discharged 200-mesh copper grids (Formvar/carbon grids, Ted Pella) for 20 s, blot-dried, and then stained three times with uranyl formate and dried. TEM imaging was performed using a Tecnai G2 Spirit BioTWIN (FEI) operating at 120-kV acceleration. Images were obtained on a TemCam-F416(R) (TVIPS) CMOS camera. To estimate the size distribution of nanoparticle–corona complexes, the size of at least 150 particles was measured by Fiji/ImageJ. Nanoparticle sizes were determined in aqueous conditions by dynamic light scattering (DLS).

**Quantification of proteins by BCA assay**. Proteins on nanoparticles were quantified by using a Pierce BCA Protein Assay Kit (Thermo Scientific). For SNPs and PsNPs, the proteins were quantified without stripping from nanoparticles. The pellet of washed nanoparticle–corona complexes was resuspended in 50 μl of PBS. About 400 μl of working reagent (copper solution) was added to the samples and

standard solutions and then incubated for 30 min at 37 °C. The samples were centrifuged at 20,000 × g for 30 min. About 200 μl of each supernatant was transferred into a 96-well plate, and the absorbance at 562 nm was measured using a Varioscan plate reader (Thermo Scientific). In order to evaluate the degree of nanoparticle interference, two control samples were performed: pristine nanoparticles in PBS and in BSA standard solutions[49]. For nanoparticles that interfere with the BCA assay, the proteins should be eluted first from nanoparticles by the protocol that was explained before in the "Azide modification of HC proteins" section and then analyzed by BCA assay.

**Eluting corona proteins from NPs**. Concentrated 5X Lane Marker reducing sample buffer (SB, Thermo Scientific, 0.3 M Tris, 5% SDS, 50% Glycerol, and 100 mM DTT) was added to nanoparticle–corona complexes to recover proteins from the NPs. The samples were heated at 95 °C for 5 min to denature and strip off proteins from NPs. DTT as a reducing agent in the sample buffer can cleave the S–S bond in the sulfo-SASD structure, which helps to cleave the clicked proteins from each other before running them on a SDS-PAGE gel. Then, the samples were centrifuged at 20,000 × g, 20 min.

**Electrophoresis analysis**. For electrophoresis analysis, the recovered proteins were diluted with PBS to adjust the sample buffer, and 40 μl of the proteins were separated on a 12% SDS-polyacrylamide gel (Bolt™ 4–12% Bis-Tris Plus Gels, 10-well) at the constant voltage of 160 V. A PageRuler Unstained Protein Ladder (Thermo Scientific) as the molecular weight standard (10–200 kDa) was also run on the gels. For CY5-labeled samples, the gels were first imaged by an Amersham Typhoon NIR laser scanner. Then, the protein bands were detected by Imperial Protein stain (Coomassie brilliant blue, Thermoscientific). The stained gels were scanned on a Bio-Rad gel documentation system. Protein quantification was performed using the plot profile tool in Fiji/ImageJ (ref). The staining intensity and run length were normalized based on the maximum values.

**LC–MS/MS analysis**. The eluted corona proteins from nanoparticles were first precipitated by using the ProteoExtract® Protein Precipitation Kit (Merck, Germany) as described in the manufacturer's protocol. The proteins were dissolved in 8 M urea and 100 mM ammonium bicarbonate with 10 mM DTT. After 30 min, adding iodoacetamide to a final concentration of 35 mM alkylated the samples. The alkylation was quenched after 30 min by adding DTT to a final concentration of 35 mM. Subsequently, the samples were diluted 5 times and digested with trypsin 1/50 (w/w) in 16 h at 37 °C. Tryptic peptides were micropurified using Empore™ SPE C18 Disks packed in 10-μl pipette tips. LC–MS/MS was performed using an EASY-nLC 1000 system (Thermo Scientific) connected to a QExactive+ Mass Spectrometer (Thermo Scientific). Peptides were trapped on a 2-cm ReproSil-Pur C18-AQ column (100-μm inner diameter, 3-μm resin, Dr. Maisch GmbH, Ammerbuch-Entringen, Germany). The peptides were separated on a 15-cm analytical column (75-μm inner diameter) packed in-house in a pulled emitter with ReproSil-Pur C18-AQ 3-μm resin. Peptides were eluted using a flow rate of 250 nl min⁻¹ and a 20-min gradient from 5 to 35% phase B (0.1% formic acid and 100% acetonitrile). The collected MS files were converted into Mascot generic format (MGF) using Proteome Discoverer (Thermo Scientific, v.24). The data were searched against the bovine proteome (uniprot.org). Database search was conducted on a local mascot search engine. The following settings were used: MS error tolerance of 10 ppm, MS/MS error tolerance of 0.1 Da, trypsin as protease, oxidation of Met as a variable modification, and carbamidomethyl as a fixed modification.

**Uni- and multivariate statistical analysis**. To test the statistical significance of differences, ANOVA analysis of the data was performed using GraphPad Prism version 8.2.1 for Windows, GraphPad Software, La Jolla, California, USA, www.graphpad.com. For cluster analysis, the copy number of corona proteins per nanoparticle was used. This approach allows a comparison of different samples without bias for large-sized proteins or the total protein input. Using this method, a protein with a higher copy number in HC+SC than in the four control HC samples is not necessarily considered an SC protein because a higher copy number could be acquired by coincidence. Therefore, the proteins classified as SC are restricted to proteins that had a consistently lower (or zero) copy number in all control samples. Clustered heatmaps were created on square root-transformed and scaled datasets using the packages gplots (ver. 3.0.1.1) and dendextend (ver. 1.9.0) in the R environment (ver. 3.5.1). For unsupervised hierarchical clustering, the distance matrix was calculated using Ward's minimum variance algorithm with the Euclidean metric.

For each corona protein identified, protein parameters, such as the grand average of hydropathy (GRAVY) scores, instability index, and isoelectric point (PI) of the proteins, were extracted using ProtParam, a tool available in the SIB ExPASY Bioinformatic Resources Portal[50]. Principal component analysis (PCA) was performed on scaled datasets using the FactoMineR (ver. 1.41)[51] and factoextra (ver. 1.0.5) packages for R to explore in a multivariate manner characteristic features of protein parameters among the corona proteins.

**Quantification of individual proteins on nanoparticles**. In order to calculate the copy number of individual protein in corona proteins on nanoparticles, three types

of measured data were employed: (1) emPAI values obtained by LC–MS/MS analysis, (2) quantified total protein mass by BCA assay, and (3) quantified nanoparticle number by reading the fluorescence of the nanoparticle. The copy number of proteins per nanoparticle was calculated by using the following expressions[52]:

$$\text{Protein mass per nanoparticle} = \text{protein content (weight \%)} \times M_{\text{total}}$$

$$= \frac{\text{emPAI} \times \text{Mw}_p}{\sum \left(\text{emPAI} \times \text{Mw}_p\right)} \times M_{\text{total}} \qquad (2)$$

$$\text{Copy number of protein per nanoparticle} = \frac{\text{protein mass per nanoparticle}}{\text{Mw}_p} \times N_A \qquad (3)$$

where protein content (weight %) is the contribution of each protein to the total adsorbed mass, $\text{Mw}_p$ is the calculated molecular weight of the protein, and $M_{\text{total}}$ is the overall mass of corona proteins per nanoparticle measured by employing BCA assay and fluorescence of nanoparticles. $N_A$ is the Avogadro constant ($6.023 \times 10^{23}$).

**Estimation of coverage of nanoparticles by corona proteins**. In order to estimate the surface coverage of nanoparticles by corona proteins, the Protein Data Bank (PDB) files of the proteins that are available were extracted from PDB: http://www.rcsb.org. Then, the structure of proteins was analyzed by PyMOL, and the minimum and maximum cross-section area of proteins were calculated. For the proteins without PDB files, by assuming the simplest shape, sphere, and this partial specific volume ($v = 0.73 \text{ cm}^3 \text{ g}^{-1}$), the volume occupied by a protein of mass $M$ in dalton and its radius were calculated as follows[53]:

$$V\left(\text{nm}^3\right) = 1.212 \times 10^{-3} \left(\frac{\text{nm}^3}{\text{Da}}\right) \times M \,(\text{Da}) \qquad (4)$$

$$R = 0.066 \times M^{1/3} \text{ (for } M \text{ in Dalton, } R \text{ in nanometer)} \qquad (5)$$

where $V$ and $R$ are the volume and radius of protein.

Since some proteins have quaternary structure and/or are in the lipoprotein structure, and it is not clear if the protein prefers to adsorb onto the nanoparticle in their natural or denatured structure, it is not possible to calculate the exact coverage of nanoparticles by proteins. On the other hand, proteins can bind to nanoparticles from their maximum and minimum cross-section area. Assuming these limitations for the most abundant proteins, different coverage values were calculated, and a range for each calculation was reported.

**Surface plasmon resonance (SPR)**. SPR measurements were made on a Biacore 3000 (Biacore AB Sweden). Gold SPR chips from SIA kit Au were cleaned with ultrasonication in acetone, ethanol, and DI water (10 min each), followed by 30 min of UV/ozone, before sputter deposition of 4-nm Ti followed by 20 nm of $SiO_2$. The $SiO_2$-coated chips were cleaned with ultrasonication in acetone, ethanol, and DI water (10 min each), followed by 30 min of UV/ozone a maximum of 1 day before use. All injections were at a rate of 5 µl min$^{-1}$ of 100 µl. First, 0.25 mg ml$^{-1}$ filtered PLL-g-PEG was injected with 10 mM HEPES (pH 7.4) as a running buffer. Next, 0.25 mg ml$^{-1}$ 70-nm $SiO_2$ NPs were injected with 10 mM NaCl as a running buffer. The running buffer was changed to 10 mM HEPES containing 100 mM NaCl (pH 7.4) (NaCl–HEPES) for all protein injections. For FBS-coated NPs, 1% FBS in NaCl–HEPES was injected prior to APOH. APOH was injected sequentially, rinsing between each injection, in concentrations 1, 15, 50, and 150 µg ml$^{-1}$ in NaCl–HEPES.

**SPR data analysis**. Linear drift corrections were applied if necessary, using an average of the background drift before and after injection. The two-dimensional fits were made on the MATLAB 2012a platform (Mathworks) using the fitting tool EVILFIT version 3 software[54,55] to determine the distribution of binding kinetics. The following input values were used for fitting the binding curves: Injection start time: Concentrations: 20, 100, 300, 1000, and 3000 nM, Start injection: $t = 0$ s, End injection: $t = 800$ s, Fit the binding phase from: $t = 2$ s, Fit the binding phase to: $t = 798$ s, Fit the dissociation phase from: $t = 1400$ s, and Fit the dissociation phase to: 2400 s.

The operator-set boundaries for the distributions were uniformly set to limit $K_D$ values in the interval from $10^{-9}$ to $10^{-3}$ M, and $K_d$ values in the interval from $10^{-5}$ to $10^0$ s$^{-1}$.

The distribution $P(k_a, K_A)$ is calculated using the discretization of the equation:

$$R_{\text{total}} = \int_{K_{a\min}}^{K_{a\max}} \int_{k_{a\min}}^{k_{a\max}} R\left(k_a, K_a, C_{\text{analyte}}, t\right) P(k_a, K_a) dk_a dK_a \qquad (6)$$

in a logarithmic grid of $(k_a, i, K_a, i)$ values with 21 grid points distributed on each axis. This was done through a global fit to association and dissociation traces at the above-mentioned analyte concentrations. Tikhonov regularization was used as described by Zhao et al.[56] at a confidence level of $P = 0.95$ to determine the most

parsimonious distribution that is consistent with the data, showing only features that are essential to fit the data.

**Cell culture**. We used THP-1 monocyte cells (a human acute monocyte leukemia cell line) obtained from the German Collection of Microorganisms and Cell Cultures (DSMZ, ACC 16) and human cerebral microvessel endothelial hCMEC/D3 cells. Both cell types were grown in RPMI 1640 medium supplemented with 10% FBS and 1% penicillin–streptomycin in a humidified 5% $CO_2$ atmosphere at 37 °C. THP-1 cells were differentiated into macrophages (hereafter "THP-1 macrophages") by incubating with 5 ng ml$^{-1}$ PMA for 48 h. The differentiated phenotype was visually inspected under an optical microscope, and then the cells were washed two times with PBS to remove PMA followed by an additional 24-h incubation in the medium without PMA[57] prior to cell experiments described below.

**Cell association of nanoparticle–corona complexes**. Cell association was assessed by a NovoCyte flow cytometer and a confocal laser-scanning microscope (CLSM, Zeiss LSM 700). For flow cytometry, 0.5 ml of hCMEC/D3 cells or THP-1 macrophages at the density of $5 \times 10^5$ cells ml$^{-1}$ were seeded in 24-well plates and exposed to the nanoparticle–corona complexes for 4 h in the RPMI media containing either 5 mg ml$^{-1}$ BSA or 10% FBS. Then, the plates were washed three times with PBS to remove free nanoparticles. The cells were then fixed by 4% paraformaldehyde for 15 min. The cells were washed three times with PBS and resuspended in 200 µl of PBS. In the flow cytometry analysis with NovoExpress software (ver. 1.4.1), at least 10,000 cells were counted. The fluorescence data are presented as median and calculated as the ratio of the median fluorescence intensity of the samples and the pristine nanoparticles in 5 mg ml$^{-1}$ BSA.

For the confocal analysis, first, the glass coverslips were coated with 50 µg ml$^{-1}$ collagen type I. For better collagen coating, the coverslips were first coated with poly-d-lysine (PDL) and then with collagen. Then, $2.5 \times 10^5$ THP-1 or hCMEC/D3 cells were seeded onto collagen precoated glass coverslips. THP-1 cells were incubated for 48 h with PMA for differentiation followed by 1 day in RPMI without PMA. hCMEC/D3 were incubated for 24 h for attachment. After the differentiation of THP-1 cells and the attachment of hCMEC/D3 cells on coverslips, the cells were exposed to the nanoparticle–corona complexes for 4 h. Then, the cells were washed and fixed with 4% paraformaldehyde. Cell nuclei were stained with 10 µg ml$^{-1}$ Hoechst 33342 (excitation: 448 nm, emission: 430–480 nm). The actin filaments were stained with 1 µg ml$^{-1}$ phalloidin–tetramethylrhodamine B isothiocyanate (excitation: 540 nm, emission: 570–573 nm). The confocal images were analyzed with ImageJ (v.1.51) and Zen (ZEISS, v.2.6).

**Reporting summary**. Further information on experimental and research design is available in the Nature Research Reporting Summary linked to this article.

## Data availability

All relevant data that support the plots within this paper and other findings of this study are available with the article and supplementary files or in the Source data file. Extra data are available from the corresponding authors upon reasonable request. Bovine proteome and Uniprot Fasta is available on https://www.uniprot.org/. The mass spectrometry proteomics data have been deposited to the ProteomeXchange Consortium via the PRIDE partner repository with the dataset identifier PXD020768[58]. The PDB format for the structure of proteins is available on Protein Data Bank (https://www.rcsb.org). The protein parameters (molecular weight, isoelectric point, GRAVY score, and instability index) were obtained from https://www.expasy.org and are provided as a Supplementary Data 1. Source data are provided with this paper.

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

## Acknowledgements

This work acknowledges funding from the FNU project DFF-4181-00473 (the role of soft interactions in complex media) and the Danish National Research Foundation center grant CellPAT (DNRF135) (H.M.-B., C.M.Z., H.E., and D.S.S.), and the grants R219-2016-327 and R324-2019-1644 from Lundbeck Foundation (Y.H.). We also acknowledge Prof. Daniel Otzen (iNANO, Aarhus University) for providing us α-synuclein protein.

## Author contributions

D.S.S., T.V.-J., and J.J.E. supervised the project. H.M.-B. and D.S.S. conceived and designed the experiments and wrote the paper. Y.H. performed the statistical and principal component analysis of the mass data and assisted with the preparation of the paper. H.M.-B., C.M.Z., and H.E. performed the experiments and analyzed the data. K.J.-M. analyzed the SPR data. C.S. performed mass spectroscopy analysis and protein identification. All authors discussed the results and commented on the paper.

## Competing interests

The authors declare no competing interests.

**Additional information**

