## [Peer Review File · Nature Communications]

Reviewers' Comments:

Reviewer #1:

Remarks to the Author:

The paper entitled „Mapping and identification of soft corona proteins at nanoparticles and their impact on cellular association“ by Mohammad-Beigi et al. is presenting a nice piece of work concerning the soft protein corona. The experiments are well done and concisely presented.

Here are my concerns/suggestions:

1. My main concern is that the functionalization of the proteins of the hard corona
 - a. includes a further manipulation step and therefore may change the protein corona (e.g. by desorption of HC proteins). The authors should discuss this in more detail.
 - b. may unfold/change the 3D structure of the proteins adsorbed. The authors should also discuss this pitfall.

This can be seen e.g. in Figure 2a and 2g when comparing HC and HC-N3
2. The authors should discuss other methods to prepare the soft corona (e.g. non-washing procedures with TEM, on-the-bead detection methods, field flow fractionation (FFF)) and should also cite literature for these aspects.
3. The authors mention in the section „Apo H shows multiple binding strengths...“ that there is a major and minor fraction of ApoH with high and low dissociation rates. Can the authors determine e.g. by mass spectrometry that these correlate to different amounts of functionalization of the HC and SC protein fraction?
4. The authors may discuss more early that uptake may be also enhanced by proteins especially when the proteins adsorb address a specific receptor of a cell line (see first paragraph of the section „SC proteins modulate cell association...“)
5. In the conclusions the authors state that they „were able to turn off their dynamic nature of dissociation...“ The authors should point out this fact very early in the paper and should state that this is a non-natural condition introduced by the experimental design.

I therefore recommend a Major Revision of the manuscript addressing above mentioned suggestions.

Reviewer #2:

Remarks to the Author:

I've read this paper with interest. Although the authors conducted huge amount of experiments, their central concept about soft corona is highly questionable (if not wrong). The steps (e.g., washing corona coated nanoparticles and functionalization of the proteins with azide groups followed by click chemistry reaction to catch soft coronas) that were explained in the paper do not capture soft coronas. In fact, modification of the hard corona with azide groups set the stage for absorption of different types of proteins rather than the soft corona.

Reviewer #3:

Remarks to the Author:

Dr Sutherland and co-workers identified a set of specific corona proteins with weak interactions at silica and polystyrene nanoparticles by using an in situ click-chemistry. They found that the majority of the captured proteins are not unique to SC but also present in the HC, suggesting that the main distinction between HC and SC is the differential binding strength of the same proteins and the same proteins can have both strong and weak interactions with nanoparticles or pre-adsorbed neighbouring proteins. The weakly interacting proteins in the SC are revealed as modulators of nanoparticle-cell association. Therefore, the weak interactions of proteins at nanoparticles should be considered when evaluating nano-bio interfaces. The click-chemistry

capture method is not a really novel approach, but the findings are quite interesting. Overall, I would recommend publication of this work after major revisions. Below are some comments to consider:

1. The proteomics data are insufficiently described. No data are provided except on a highly aggregated level. The entire dataset should be presented (in the manuscript or supplementary information). Please refer to MCP reporting guidelines for proper reporting of proteomic datasets. Additionally, I highly recommend submitting the datasets (raw data and search results) to a public repository (i.e. Proteome X change) to allow other researchers to access the data.

2. A key finding here is the influence of corona on cell association. However, the corona protein is of calf origin and the cells are of human origin. How do species-specific effects contribute to the observed results? The questions should be addressed experimentally: whether similar results can be observed if nanoparticles are incubated in human serum and then exposed to human macrophage-like cells.

3. The aggregation state of the corona complexes used in this study may provide an additional variable influencing the observed cell results. Using an appropriate method, authors should assess the level of aggregation of their materials both after serum exposure and in cell culture media. A distribution of this quantity should be discussed.

4. Nanoparticles with charged surface groups or hydrophobic surface have more stable interactions with proteins than nanoparticles with neutral surface charges and hydrophilic surface. Studies on representative nanoparticles with different surface properties are highly suggested to show the general nature of the click-chemistry capture method.

Responses to the Reviewers Comments and Questions

All changes in the revised version are marked **in red**

Reviewer#1

The paper entitled “Mapping and identification of soft corona proteins at nanoparticles and their impact on cellular association” by Mohammad-Beigi et al. is presenting a nice piece of work concerning the soft protein corona. The experiments are well done and concisely presented. Here are my concerns/suggestions:

1. My main concern is that the functionalization of the proteins of the hard corona

a. includes a further manipulation step and therefore may change the protein corona (e.g. by desorption of HC proteins). The authors should discuss this in more detail.

b. may unfold/change the 3D structure of the proteins adsorbed. The authors should also discuss this pitfall. This can be seen e.g. in Figure 2a and 2g when comparing HC and HC-N3

Response: We appreciate the reviewer’s positive evaluation of the work. We understand the reviewer’s concern regarding the further manipulation steps to functionalize proteins which are a necessary part of our approach and have added new text within the manuscript to discuss this. There is a risk that the additional centrifugation steps to separate hard corona during functionalization steps may desorb some additional proteins from the HC proteins. As with all labelling approaches, there of course the possibility for modulations of specific protein behaviours and 3D structure if the labelling is extensive. In terms of the amount of protein binding we have quantified the amount of hard corona proteins before and after N₃ labelling with the BCA assay and did not see any significant difference in the amount of protein bound at nanoparticles so we believe that any additional desorbed amount must be small. With respect to the issue of labelling, we have chosen a relatively low level of labelling (we estimate approx. 1.2 groups per protein for a protein with an average size around that of BSA) which minimizes the effect of any labelling. With respect to the protein structure it is technically difficult to study directly, conformational changes in a mixed population of different proteins plus the protein conformation is likely already altered for surface bound proteins compared to unbound proteins. We are however able to measure the result of the competitive binding of proteins to the nanoparticle surfaces which we considered as relevant to gauge if there have been significant changes in for example protein conformation. In the proteomics analysis (in for example figure 2a and 2g that that is referred to by the reviewer), we do observe some small quantitative differences between the labelled and unlabelled HC

samples (HC and HC-N₃), but our cluster analysis show that they are in the same cluster and all controls are clearly separated from HC+SC samples in the column dendrogram.

We discussed this and added the following to the Discussion part of the revised manuscript:

PAGE.5. “It should be mentioned to avoid the potential for changes of the overall protein interactions with nanoparticles, e.g. highly modified BSA¹, we labelled the HC proteins with N₃ after formation on NPs and with a relatively low level of labelling. The further centrifugation and manipulation steps applied in this method such as N₃ modification did not significantly desorb HC proteins from nanoparticles; however, we believe that slightly effect of these modifications on the SC profile can potentially occur.”

2. The authors should discuss other methods to prepare the soft corona (e.g. non-washing procedures with TEM, on-the-bead detection methods, field flow fractionation (FFF)) and should also cite literature for these aspects.

Response: We agree we should have included such a discussion of other methods and have now added it to in the present submission. We added the following to the Introduction:

PAGE.2 and 3. “Recent work has developed approaches to quantify SC protein binding and address the potential of soft interactions to modulate toxicity by localized sulphidation at the surface of silver nanoparticles². Several methods such as non-washing procedures³ or multi-step centrifugation⁴ are proposed to retain a larger fraction of the hard corona proteins for identification during separation albeit still after long times. Asymmetric flow field-flow fractionation (A4f)⁵, and surface plasmon resonance coupled with mass spectroscopy⁶ have been applied to PEGylated nanoparticles to identify weakly protein binding proteins in stealth systems. However for the rapidly exchanging SC proteins, several key open questions remain, including whether SC proteins are different from HC proteins and if there is a role for SC proteins in determining cellular interactions.”

In the Discussion we added:

PAGE.5 “Weber et al⁵ by using an asymmetric flow field-flow fractionation (A4f) separation technique identified SC proteins on surfactant/polystyrene NP complexes and human serum albumin (HSA) was considered to be associated as a SC protein; however, similar to our work, they also showed that the SC proteins are mainly present in the HC. Kari et al found enrichment of stealth mediating proteins towards phagocytes that were enriched as SC proteins on liposomes immobilized on an SPR sensor⁶.”

PAGE.8 “Weber et al ⁵ previously showed that only the hard protein corona on surfactant/polystyrene NP complexes directly influenced the uptake of NPs by Hela cells and SC did not alter the biological behaviour.”

3. The authors mention in the section „Apo H shows multiple binding strengths...“ that there is a major and minor fraction of ApoH with high and low dissociation rates. Can the authors determine e.g. by mass spectrometry that these correlate to different amounts of functionalization of the HC and SC protein fraction?

Response: We should clarify that the SPR measurements were done with label-free HC and SC proteins and the different binding states of APO H were identified by fitting the SPR data and absorption of APO H on HC. We clarified this in the revised version of the manuscripts and explained more about the competition study and setup. We added the following to the results:

PAGE.6 “Using different eluting solutions (reducing and non-reducing sample buffers and DTT solution) confirmed that a reducing agent is necessary to elute the majority of α -Syn from the nanoparticles. This confirms that α -Syn is captured mainly through the click reaction and DTT can reduce the disulphide bridge in the Sulpho-SASD structure to release α -Syn.”

PAGE.6 “To study the soft binding property of APO H without click reactions, we utilised an immobilised-nanoparticle SPR setup to quantify the interaction of free-label APO H with FBS proteins deposited onto SNPs 37 (Fig. 3g). In this setup, SNPs were first immobilized on repellent PLL-g-PEG, as a protein resistant polymer, which reduces unspecific protein binding directly on the SPR biosensor. Then, protein corona was formed by injecting 1 % FBS onto the immobilized NPs.”

PAGE.7 “A rapid drop of up to 60 % of the adsorbed APO H on FBS-coated SNPs was seen in 50 μ g/ml APO H, while this drop is only 6 % for adsorption of FBS on SNPs, indicating that the FBS proteins are tightly adsorbed to the NPs (supplementary Fig. 15).”

4. The authors may discuss more early that uptake may be also enhanced by proteins especially when the proteins adsorb address a specific receptor of a cell line (see first paragraph of the section „SC proteins modulate cell association...“)

Response: Thanks for the reviewer for pointing this out. We added the following to the introduction:

PAGE.2 “The formation of protein coronas leads to two main consequences that determine how well the nanoparticles associate with cells. Under a serum-free condition, pristine nanoparticles

spontaneously bind to cell membranes in a non-specific manner lowering their surface energy and protein coronas, in general, reduce this non-specific interaction as less nanoparticle surface is exposed⁷. In parallel, nanoparticle-bound proteins provide the potential for specific interactions during cell association (CA), including receptor-mediated membrane adhesion and subsequent uptake^{21,33} and contribute to the resultant biomolecular corona defined biological identity of the nanoparticles^{8,9}. Therefore, to predict the biological behaviour of nanoparticles, it is essential to have a combined understanding of the composition and structure of protein corona.

Kinetic evaluation of protein corona formation and identification of the proteins forming the corona have become active research topics aiming to understand the particokinetics, cellular interactions, and mechanisms of nanoparticle toxicity²”

5. In the conclusions the authors state that they „were able to turn off their dynamic nature of dissociation...” The authors should point out this fact very early in the paper and should state that this is a non-natural condition introduced by the experimental design.

Response: We agree that this should be highlighted early in the paper and have now added the following to the introduction:

PAGE.2 “Moreover, turning off the dynamic nature of dissociation, which is the modulation of real condition for cell studies, provided us the possibility to study the effect of the dynamic nature of SC proteins on cell association.”

And the following to the discussion:

PAGE.7 “This is a modulation of the real condition for nanoparticle-cell interaction studies but it provides a means for studying the effect of the dynamic exchange of soft corona proteins on cell associations.”

I therefore recommend a Major Revision of the manuscript addressing above mentioned suggestions.

Reviewer#2

I've read this paper with interest. Although the authors conducted huge amount of experiments, their central concept about soft corona is highly questionable (if not wrong). The steps (e.g., washing corona coated nanoparticles and functionalization of the proteins with azide groups followed by click chemistry reaction to catch soft coronas) that were explained in the paper do not capture soft coronas. In fact,

modification of the hard corona with azide groups set the stage for absorption of different types of proteins rather than the soft corona.

Response: The reviewer has raised doubts about the potential for our approach we believe because the click chemistry approach has been widely used to capture bulk proteins at nanoparticles¹⁰. We should have discussed the different situation in our approach more clearly to explain why soft proteins coronas can be captured and identified. First we address the effect of labelling on over protein interactions. We appreciate that labelling provides the potential for changes of the overall protein, interactions with nanoparticles, e.g. highly modified BSA¹. We had considered all these potentials before designing our method and we would be happy to clarify it.

We designed three sets of experiments to prove that the profile of SC proteins captured by the click reaction was not affected by the labelling and that the proteins captured were not from the bulk:

1- Labelling efficiency

1-1 We labelled HC proteins with N₃ after their formation on nanoparticles to avoid the effect of N₃ on the HC profile and formation.

1-2 Previous studies that demonstrated the effect of labelling on the protein corona formation, used proteins that were highly labelled which for example gave the proteins significantly different zeta potential values¹. In our study, we have chosen a relatively low level of labelling (approx.. 1.19 N₃ per protein for a protein with an average size around that of BSA, supplementary Fig.S1 a). By using DBCO-CY5 we proved, that all protein types are labelled with N₃ (supplementary Fig. S1 b and c) but the zeta potential measurement confirmed that this level of labelling did not change the total charge of HC significantly (Table.1, -20±4.1 mv vs -25±2.4 mv for SNPs and -26.2.3 vs -25±3.1 for PsNPs).

2- Proteomics data

2-1 The proteomics data show that for SNPs, 20 proteins were considered as SC proteins among the total of 80 proteins identified by LC-MS/MS, and only 4 out of the 20 SC proteins were uniquely captured after the click reaction, while the others were found in the HC controls to some extent (Fig. 2b). This indicates that the click reaction does not bind random new proteins from the bulk phase.

2-2 The profile of enriched SC proteins is not similar to the profile of bulk proteins. The absence of highly abundant serum proteins such as albumin in the SC cluster shows that

our click chemistry method used in a competitive situation does not capture, substantially, bulk proteins and instead captures proteins that are resident at the surface through a weak interaction with HC proteins and /or NPs's surface and not proteins directly from the bulk. Published work in the literature where bulk proteins are captured at nanoparticles are made from pure protein solutions of high concentration and have no competition from other (soft corona) proteins allowing the bulk protein to interact. Notably, while there was a 30% overlap of HC cluster proteins between SNPs and PsNPs, however only 7% of SC cluster proteins were common to both nanoparticle types (Supplementary Fig. S12). This also shows that dependent on the type of HC and nanoparticles, different SC proteins are captured.

2-3 The number-weighted averages of the overall protein characteristics (calculated molecular mass, isoelectric point, instability index, and gravy index) of the HC controls and the HC+SC sample did not show a particular propensity indicating that the click reaction itself does not preferentially capture proteins with a specific parameter (Supplementary Fig. S11).

3- Competition study

A series of competition studies were performed using APO H, BSA, and FBS to confirm that the click reaction only captures the proteins, which in a competitive situation, could compete with bulk proteins and stay close enough to the HC through a weak interaction with HC proteins and/or the NP's surface.

3-1 APO H as the most enriched SC protein on SNPs was labelled with DBCO and BSA was selected as the direct competitor since it is an HC cluster protein and has a higher affinity for the surface of SNPs than APO H. While APO H showed weaker binding to uncoated SNPs in the presence of the competitor BSA (Fig.3 a,b), APO H was effectively enriched by the click reaction on the HC pre-assemble on SNPs, outcompeting BSA with negligible influence of the concentration ratio between the two (Fig. 3 c,d). T shows that APO H can outcompete bulk proteins for the HC coated nanoparticle surface even if that bulk protein has a stronger interaction with the nanoparticle core.

3-2 While BSA was captured on HC on SNPs when in pure solution, the low competitiveness of BSA for soft interactions with SNPs was supported by the reduced capture of BSA via the click chemistry reaction, despite its high abundance, in FBS (Fig.3 e,f).

We have expanded the revised manuscript to better describe our approach in response to the reviewer's comments. We added the following to the text:

PAGE.3 "Sulpho-SASD and DBCO-Sulpho-NHS were used for the modification of proteins to perform the click chemistry reaction described in Fig.1a. The modification occurs through the reaction between Sulpho-NHS moieties on the crosslinkers with primary amines on proteins. None of the azide or DBCO reactive groups on these heterobifunctional crosslinkers react with any of the functional groups on proteins, which avoid crosslinking of HC or SC proteins with other HC or SC ones. Moreover, Sulpho-SASD contains a dithiol, which provides a possibility to cleave the covalent bond between proteins by using reducing agents for analysis."

PAGE.4 "It should be mentioned to avoid the potential for changes of the overall protein interactions with nanoparticles, e.g. highly modified BSA¹, we labelled the HC proteins with N₃ after formation on NPs and with a relatively low level of labelling. The further centrifugation and manipulation steps applied in this method such as N₃ modification did not significantly desorb HC proteins from nanoparticles; however, we believe that slightly effect of modifications on the SC profile is unavoidable.

We first calculated the copy number of each identified protein per nanoparticle following quantitation of the total protein mass (by BCA assays), nanoparticle mass (by fluorimetry) and emPAI-based relative mass percentages of proteins identified in LC-MS/MS (see Supplementary Information for details). This allows a comparison of different samples without bias for large-sized proteins or the total protein input."

PAGE.4 "In this approach, having a higher copy number than in the four control HC samples is not automatically considered to be indicative of an SC protein because a higher copy number may also be acquired by random variation. Therefore, the SC protein cluster we identified here is restricted to proteins that had a consistently lower (or zero) copy number in all of the four control samples without a large variation among them."

PAGE.4 "This can also be explained by the fact that the top 5 abundant proteins, which account for >50% of the total, remained the same even after the click reaction, whereas SC cluster proteins were ranked higher than before (e.g. 5.7-fold increase in the abundance of APO H; Table 2, "SNPs" and Fig. 2b)."

PAGE.5 "This classification is visualized in Fig. 2d-e, where only the SC cluster proteins are displayed but with a different colour code for each SC type. The total copy number of the SC cluster proteins increased

~2-fold after the click reaction with Type 3 SC proteins representing the major SC fraction which were effectively captured (Fig. 2d) and thus overall increased in the copy number (Fig. 2e).”

PAGE 5. “It should be mentioned that despite almost the same negative charge on both SNPs and PsNPs, the classification of corona proteins on SNPs and PsNPs was different (Supplementary Figs. S10), as thoroughly discussed in the previous studies³³. This shows that other than electrostatic interactions between proteins and NPs with opposite charges, there can be electrostatic interactions between the exposed part of a denatured protein with the same charge, and nanoparticles, or other interactions such as hydrophobic interactions, as PsNPs are more hydrophobic.”

PAGE 5. “Weber et al⁵ by using an asymmetric flow field-flow fractionation (A4f) separation technique identified SC proteins on surfactant/polystyrene NP and human serum albumin (HSA) was considered to be associated as a SC protein; however, similar to our work, they also showed that the SC proteins are mainly present also in the HC. Kari et al⁶ found enrichment of stealth mediating proteins towards phagocytes that were enriched as SC proteins on liposomes immobilized on an SPR sensor.”

Reviewer#3

*Dr Sutherland and co-workers identified a set of specific corona proteins with weak interactions at silica and polystyrene nanoparticles by using an in situ click-chemistry. They found that the majority of the captured proteins are not unique to SC but also present in the HC, suggesting that the main distinction between HC and SC is the differential binding strength of the same proteins and the same proteins can have both strong and weak interactions with nanoparticles or pre-adsorbed neighbouring proteins. The weakly interacting proteins in the SC are revealed as modulators of nanoparticle-cell association. Therefore, the weak interactions of proteins at nanoparticles should be considered when evaluating nano-bio interfaces. The click-chemistry capture method is not a really novel approach, **but the findings are quite interesting. Overall, I would recommend publication of this work after major revisions.** Below are some comments to consider:*

1. The proteomics data are insufficiently described. No data are provided except on a highly aggregated level. The entire dataset should be presented (in the manuscript or supplementary information). Please refer to MCP reporting guidelines for proper reporting of proteomic datasets. Additionally, I highly recommend submitting the datasets (raw data and search results) to a public repository (i.e. Proteome X change) to allow other researchers to access the data.

Response: We agree that the proteomics data should have been better presented. Now, the full proteomics data containing the proteins accession number, their parameters and numbers/NPs are submitted in an excel file as supplementary information. We also added more explanation in the proteomics discussion part of the revised manuscript, highlighted in red.

2. A key finding here is the influence of corona on cell association. However, the corona protein is of calf origin and the cells are of human origin. How do species-specific effects contribute to the observed results? The questions should be addressed experimentally: whether similar results can be observed if nanoparticles are incubated in human serum and then exposed to human macrophage-like cells.

Response: We appreciate that the origin of corona protein can affect the cell association of nanoparticles. For the cell study, we chose to work with FBS proteins at the nanoparticle to mimic the normal protein environment for in vitro cell studies. The cells we chose to work with are a cell line routinely grown in and conditioned to FBS containing media. This system is an *in vitro* model system for study and we believe that comparison of FBS proteins in this admittedly human cell system (but grown in FBS media) provides the best standardized experimental system. This model system has been used extensively for in vitro cytotoxicity studies of different nanoparticle types. While we could in principle form protein corona in human serum and then expose them to the human cell line in for example RPMI medium + 10 % human source serum, we believe that the widespread use of FBS in *in vitro* culture models used for cytotoxicity even with human cell lines (e.g. Ref. 20 in the main text : “Yan, Y. *et al.* Differential roles of the protein corona in the cellular uptake of nanoporous polymer particles by monocyte and macrophage cell lines. *ACS nano* **7**, 10960–10970 (2013).”), makes this a better choice for comparison in relation to previous literature. We have however now clearly highlighted this choice of serum in the manuscript.

Text added

PAGE 7. “ It should be noted that we use bovine origin serum with these human-derived but FBS-conditioned cell lines.”

3. The aggregation state of the corona complexes used in this study may provide an additional variable influencing the observed cell results. Using an appropriate method, authors should assess the level of aggregation of their materials both after serum exposure and in cell culture media. A distribution of this quantity should be discussed.

Response: We thank the reviewer for pointing out this omission. The size distribution of particles in RPMI+10 % FBS medium measured by DLS is now added to the supplementary Table. S4 which is included here for reference:

Supplementary Table S4. Size distribution of nanoparticle-corona complexes in cell culture medium (RPMI+10 % FBS)

	nanoparticle-corona complexes	DLS hydrodynamic diameter ± SD (nm) (PDI)		nanoparticle-corona complexes	DLS hydrodynamic diameter ± SD (nm) (PDI)
SNPs	pristine	112 ±6.3 (0.10)	PsNPs	pristine	145±7.4 (0.09)
	HC	138±7.7 (0.11)		HC	165±8.7 (0.1)
	HC+SC	161±9.4 (0.21)		HC+SC	203±10.1 (0.33)

Supplementary Table S4. The average size of nanoparticle-corona complexes was determined using DLS Data in cell culture medium (RPMI+10 % FBS). Data shown correspond to mean ± sd. of three independent experiments (n=3).

We also added the following to the Result part:

PAGE 8. “To evaluate the level of aggregation of the NPs in the cell medium (RPMI+10 % FBS) which can influence the cell association of NPs, the size distribution of nanoparticle-corona complexes was determined using DLS (supplementary Table S4). The results showed that the particles in the cell medium were still stable against aggregation. We expect that the accumulation of particles observed in the confocal images occurred after cell association.”

4. Nanoparticles with charged surface groups or hydrophobic surface have more stable interactions with proteins than nanoparticles with neutral surface charges and hydrophilic surface. Studies on representative nanoparticles with different surface properties are highly suggested to show the general nature of the click-chemistry capture method.

Response: We had already considered the reviewer’s concern regarding the generality of the nature of the click-chemistry capture method and we have tested this method on negatively charged hydrophilic silica nanoparticles (SNPs, 70 nm) and hydrophobic-character carboxyl-modified polystyrene nanoparticles (PsNPs, 100 nm).

Now, we have added new data illustrated the applicability of this method to other types of nanoparticles with different surface chemistry and charge and we observed comparable BCA and SDS-

PAGE results for amine-modified SNPs (SANPs) with both positive and negative charge groups, and highly charged carboxyl-modified SNPs (SCNPs).

The DLS and zeta potential of SANPs and SCNPs are summarized in the new supplementary Table. S2 (shown in the following) and their SDS-PAGE and quantification data are shown in the new supplementary Fig. S5 in the revised version of the manuscript.

Supplementary Table S2. Characterization of nanoparticle-corona complexes in buffer

	nanoparticle-corona complexes	Zeta potential ± SD (mV)	DLS hydrodynamic diameter ± SD (nm) (PDI)		nanoparticle-corona complexes	Zeta potential ± SD (mV)	DLS hydrodynamic diameter ± SD (nm) (PDI)
SANPs	pristine	-14±2.1	79±2.8 (0.03)	SCNPs	pristine	-38±2.4	82±3.5 (0.02)
	HC	-13±1.8	101±5.3 (0.05)		HC	-33±2.1	109±4.2 (0.04)
	HC-N ₃	-13±1.3	115±10.2 (0.11)		HC-N ₃	-36±1.9	119±5.8 (0.09)
	D Ctrl	-14±2.5	117±7.8 (0.14)		D Ctrl	-30±0.9	123±4.1 (0.10)
	N ₃ Ctrl	-16±1.7	127±6.3 (0.16)		N ₃ Ctrl	-29±1.3	129±7.8 (0.09)
	HC+SC	-17±2.9	148±9.1 (0.18)		HC+SC	-31±1.8	156±8.3 (0.12)

Supplementary Table S2. The average size of nanoparticle-corona complexes was determined using DLS and the zeta potential measurement data processing was done by using Smoluchowski model. Zeta potential measurement was done in 10 mM sodium phosphate buffer, pH 7.4, containing 10 mM NaCl. Data shown correspond to mean ± sd. of three independent experiments (n=3).

The following is added to the results part of the revised version of the manuscript:

PAGE 4. “Illustrating the applicability of this method to other types of nanoparticles with different surface chemistry and charge, we observed comparable BCA and SDS-PAGE results for mixed charge amine-modified SNPs (SANPs, 75 nm), highly charged carboxyl modified SNPs (SCNPs, 75 nm), and PsNPs (Fig. 1f and Supplementary Fig. S5).”

References:

1. Treuel, L. *et al.* Impact of protein modification on the protein corona on nanoparticles and nanoparticle-cell interactions. *ACS Nano* **8**, 503–513 (2014).
2. Micluas, T. *et al.* Dynamic protein coronas revealed as a modulator of silver nanoparticle sulphidation in vitro. *Nat Commun* **7**, 11770 (2016).

3. Kokkinopoulou, M., Simon, J., Landfester, K., Mailänder, V. & Lieberwirth, I. Visualization of the protein corona: towards a biomolecular understanding of nanoparticle-cell-interactions. *Nanoscale* **9**, 8858–8870 (2017).
4. Bonvin, D., Chiappe, D., Moniatte, M., Hofmann, H. & Ebersold, M. M. Methods of protein corona isolation for magnetic nanoparticles. *Analyst* **142**, 3805–3815 (2017).
5. Weber, C., Simon, J., Mailänder, V., Morsbach, S. & Landfester, K. Preservation of the soft protein corona in distinct flow allows identification of weakly bound proteins. *Acta biomaterialia* **76**, 217–224 (2018).
6. Kari, O. K. et al. In situ analysis of liposome hard and soft protein corona structure and composition in a single label-free workflow. *Nanoscale* (2020).
7. Lesniak, A. et al. Effects of the presence or absence of a protein corona on silica nanoparticle uptake and impact on cells. *ACS Nano* **6**, 5845–5857 (2012).
8. Cedervall, T. et al. Understanding the nanoparticle-protein corona using methods to quantify exchange rates and affinities of proteins for nanoparticles. *P Natl A Sci* **104**, 2050–2055 (2007).
9. Patel, H. Serum opsonins and liposomes: their interaction and opsonophagocytosis. *Crit Rev Ther Drug* **9**, 39–90 (1992).
10. Liu, X. et al. Rapid conjugation of nanoparticles, proteins and siRNAs to microbubbles by strain-promoted click chemistry for ultrasound imaging and drug delivery. *Polym Chem* **10**, 705–717 (2019).

Reviewers' Comments:

Reviewer #1:

Remarks to the Author:

The authors have addressed all my concerns in full and I recommend to publish the paper as is.

Reviewer #2:

Remarks to the Author:

This reviewer appreciates the authors' efforts in better clarifying their claims. The main concern of this reviewer (i.e., chemical modification of hard corona will change the formation of soft corona), however, remained intact. The use of "semi-soft" corona might overcome, at least in part, this major issue of the manuscript.

The authors are encouraged to acknowledge previous works on developing suitable approaches to analyze soft corona at the surface of nanoparticles. The readers may also benefit from discussion (in terms of reproducibility of the proteomic outcomes) regarding the critical role of the choice of protein source (e.g., human plasma, serum, etc; and their health spectrum) on the profiles of soft and hard coronas.

Reviewer #3:

Remarks to the Author:

This is a significantly revised manuscript with more data. This reviewer feels that the authors tried to address a big question with too many variables. I recommend its rejection to Nature Commun.

For some remaining questions/suggestions:

1. The authors may need a more detailed discussion about cell association of corona complexes in phagocytic and non-phagocytic cells. For example, the interaction between cell membrane and corona complexes.
2. Inorganic nanoparticles (such as quantum dots) smaller than 10 nanometers in diameter similar to proteins found in serum may affect analytical separation approach proposed by authors. Furthermore, the metallic cores of metallic NPs and metal oxide NPs may influence the analysis methods of click-chemistry reaction.
3. The formation of the corona is subject to crowding/cooperative effects. With the increasing combination of substrate and macromolecule, the binding of later ligands will be affected due to the presence of ligands. The crowding effects will lead to instability in the functionalization of HC proteins with azide.

Responses to the Reviewers Comments and Questions

All changes in the revised version are marked **in red**

Reviewer #1

The authors have addressed all my concerns in full and I recommend to publish the paper as is.

Reviewer #2

1) This reviewer appreciates the authors' efforts in better clarifying their claims. The main concern of this reviewer (i.e., chemical modification of hard corona will change the formation of soft corona), however, remained intact. The use of "semi-soft" corona might overcome, at least in part, this major issue of the manuscript.

Response: We appreciate the reviewers' concerns. The fact that we needed to chemically modify the proteins to capture interacting counterparts imply there can be such concerns. In the previous revision, we attempted to make a clearer case for our conclusions, highlighting that the chemical modification of the HC is optimised to a low level (one modified amino acid residue per protein), not biased to specific proteins, and also not capturing random bulk proteins. We also showed how the binding strength of individual proteins could be studied without any chemical modification, once the protein was identified, by using SPR. We would also point out that our experimental result was that we did not capture a large number of new proteins (although we did identify some) nor did we capture a random fraction of the bulk proteins. This suggests that our chemical modification approach does not lead to induced binding of new proteins which results in false identification of SC proteins. In this way, our approach can be considered to be rather conservative with a low risk of false identification while providing a solid proof of protein-protein interactions that occur in the proximity of the nanoparticle surface. What we show here by providing evidence that the majority of proteins identified were indeed proteins also belonging to hard corona is in good agreement with how the hard corona has been defined to date. Washing of the corona by centrifugation is the most practised isolation techniques, and our belief is that free and weakly interacting proteins are removed during the process. It is the weakly interacting proteins that are considered as the SC proteins; our results do support this view by arguing that weakly interacting proteins can also have strong interactions with the nanoparticle surface and persist during washing. We were able to immobilise the weakly interacting proteins by anchoring to the strongly interacting proteins without signature of capturing random bulk proteins such as serum albumin.

The reviewer has also raised another important issue on the nonclemanture of the proteins which we study. Within the paper we have presented an approach to the capture and identification of proteins that are not strongly enough attached at the nanoparticle to remain bound after separation of the nanoparticle from the protein media (e.g. exemplified here by centrifugation). We have used the broad term 'soft corona proteins' to refer to these proteins because this terminology is present in the literature. We have been clear about limitations of our approach that make it likely that we do not capture all the proteins of this type. The term soft corona is somewhat of a simplification and encompasses different types of proteins which may bind weakly but non-specifically to the particle surface or specifically to other proteins at the particle. We had chosen to call them 'soft corona proteins' in the title and specifically not 'the soft corona' since we did not want to claim to have captured the complete corona. We observe differences in strength of interaction in terms of the relevalive amounts of proteins (or absence of proteins) in the hard protein corona versus the captured soft protein corona and used these to classify three forms of soft corona proteins (SC1, SC2 and SC3 where SC3 includes the proteins that were not found in the hard corona at all). The reviewer has suggested that we use a term semi-soft. Clearly our observation is that the majority of the identified proteins are also present at some level in the hard corona. However the observed proteins are not themselves part of the hard corona and we point out that we observe proteins of different apparent binding strength and some that are not present in the hard corona at all. We also point out that do not claim to have identified the whole soft corona, but can see that there could be misunderstandings of what we meant by our terminology, but feel that semi-soft has the same issues of potential misinterpretation. We have defined now within the text what we mean by soft-corona proteins.

PAGE.3 "Here, by developing an experimental approach based on click chemistry, we captured weakly interacting proteins along with HC proteins for mass spectrometry-based compositional profiling and identify proteins that are either new or with increased amount compared to HC layers as soft corona proteins. We find that the majority of the identified SC proteins were not unique to SC but were also present in the HC representing different binding strength states of the same proteins."

We have modified the title of the paper to replace 'soft-corona proteins' with 'weakly interacting corona proteins' since we do not have space in the title to define exactly what we meant (in order to avoid any misunderstandings before readers progressed further into the work). The new title is:

"Mapping and identification of weakly interacting corona proteins at nanoparticles and their ignored impact on cellular association"

2) *The authors are encouraged to acknowledge previous works on developing suitable approaches to analyze soft corona at the surface of nanoparticles.*

Response: We agree that a discussion of other methods should be done in the manuscript. We had in fact already added such a discussion in the previous version of revised manuscripts as requested by reviewer 1. We have now extended it in the current revision:

Previous additions:

PAGE 3. “Recent work has developed approaches to quantify SC protein binding and address the potential of soft interactions to modulate toxicity by localized sulphidation at the surface of silver nanoparticles ². Several methods such as non-washing procedures ³ or multi-step centrifugation⁴ are proposed to retain a larger fraction of the hard corona proteins for identification during separation albeit still after long times. Asymmetric flow field-flow fractionation (A4f) ⁵, and surface plasmon resonance coupled with mass spectroscopy ⁶ have been applied to PEGylated nanoparticles to identify weakly protein binding proteins in stealth systems. However for the rapidly exchanging SC proteins, several key open questions remain, including whether SC proteins are different from HC proteins and if there is a role for SC proteins in determining cellular interactions.”

PAGE.5 “Weber et al ⁵ by using an asymmetric flow field-flow fractionation (A4f) separation technique identified SC proteins on surfactant/polystyrene NP complexes and human serum albumin (HSA) was considered to be associated as a SC protein; however, similar to our work, they also showed that the SC proteins are mainly present in the HC. Kari et al found enrichment of stealth mediating proteins towards phagocytes that were enriched as SC proteins on liposomes immobilized on an SPR sensor ⁶.”

PAGE.8 “Weber et al ⁵ previously showed that only the hard protein corona on surfactant/polystyrene NP complexes directly influenced the uptake of NPs by Hela cells and SC did not alter the biological behaviour.”

New version:

PAGE.2 “Recent work has developed approaches to quantify SC protein binding and address the potential of soft interactions to modulate toxicity by localized sulphidation at the surface of silver nanoparticles ¹⁴; Several methods such as centrifugation-based separation techniques together with proteomic characterization ²³ or multi-step centrifugation²⁴ are proposed to retain a larger fraction of the hard corona proteins for identification during separation albeit still after long times. In the centrifugation-based separation technique, by using transmission electron microscopy, it is shown that

protein corona is an undefined loose network of proteins; however, in that method there is a risk to capture bulk proteins between nanoparticles during the centrifugation. Asymmetric flow field-flow fractionation (A4f) ²⁵ and surface plasmon resonance coupled with mass spectroscopy ²⁶ have been applied to PEGylated nanoparticles to identify weakly protein binding proteins in stealth systems. While in the SPR method, SC and HC proteins are identified in a label free method and the SC proteins are found as the stealth component of the biological identity, it is tested only on liposomes. For the rapidly exchanging SC proteins, several key open questions remain, including whether SC proteins are different from HC proteins and if there is a role for SC proteins in determining cellular interactions.”

The following comparison of the effect of different soft corona proteins on cellular association of nanoparticles is also added to the results part:

PAGE.9 “While Weber et al ²⁵ showed that only the hard protein corona on surfactant/polystyrene NP directly influenced the uptake of NPs by HeLa cells and SC did not alter the biological behaviour, Kokkinopoulou et al showed that soft corona reduces the cellular uptake of nanoparticles in comparison to hard corona coated nanoparticles ²³. Kari et al also suggested that soft corona of loosely interaction proteins on liposomes contributes to the stealth properties as a component of the biological identity ²⁶ ”

3) The readers may also benefit from discussion (in terms of reproducibility of the proteomic outcomes) regarding the critical role of the choice of protein source (e.g., human plasma, serum, etc; and their health spectrum) on the profiles of soft and hard coronas.

Response: We appreciate that the origin of corona protein can affect the proteomic outcomes and cell association of nanoparticles. We clarified the critical role of protein source in the revised version of the manuscript. The following was added to the results:

PAGE.6 “Here, we chose FBS as a culturing condition, which is routinely used for *in vitro* cell association studies. We expect to capture different HC and SC proteins by using different protein sources as it is shown before that even slight variation in the composition can substantially change the profile and amount of the hard protein corona on nanoparticles. Ezzat et al ³⁶ showed that different protein sources such as human plasma, human bronchoalveolar lavage fluid, and fetal serum proteins not only affect the composition of protein corona on viruses but also affect viral infectivity and immune cell activation. Hajipour et al also ³⁷ confirmed that protein pattern on silica and polystyrene nanoparticles differed both in terms of composition and amount with the various disease.”

Reviewer #3

This is a significantly revised manuscript with more data. This reviewer feels that the authors tried to address a big question with too many variables. I recommend its rejection to Nature Commun.

For some remaining questions/suggestions:

1. The authors may need a more detailed discussion about cell association of corona complexes in phagocytic and non-phagocytic cells. For example, the interaction between cell membrane and corona complexes.

Response: We appreciate that the cell association of corona complexes to phagocytic and non-phagocytic cells is different and the discussion on this topic is now extended in the revised version of the manuscript. This was added to the results:

PAGE.2 “Cell association is also related to the differences between cell types (e.g., phagocytic vs non-phagocytic, cancer vs normal cells, and monocytes vs macrophages). THP-1 cell lines exhibit enhanced expression of macrophage surface markers as compared to primary monocytes and macrophages³⁹. In this study, we used PMA to induce differentiation of THP-1 cells to a macrophage phenotype to represent a cell type expressing a wider variety of receptors for phagocytosis as in primary macrophages, although some differences in the gene expression profile exist between the two (e.g., a higher expression level of scavenger receptor A⁴¹). Endothelial cells such as hCMEC/D3 cells, on the other hand, are active in endocytosis facilitating cross-cellular transport of nutrients and other biomolecules⁴⁸, notably including FcRn receptors for BSA⁴². ”

PAGE.9 “While Weber et al²⁵ showed that only the hard protein corona on surfactant/polystyrene NP directly influenced the uptake of NPs by HeLa cells and SC did not alter the biological behaviour, Kokkinopoulou et al showed that soft corona reduces the cellular uptake of nanoparticles in comparison to hard corona coated nanoparticles²³. Kari et al also suggested that soft corona of loosely interaction proteins on liposomes contributes to the stealth properties as a component of the biological identity²⁶ ”

2. Inorganic nanoparticles (such as quantum dots) smaller than 10 nanometers in diameter similar to proteins found in serum may affect analytical separation approach proposed by authors. Furthermore, the metallic cores of metallic NPs and metal oxide NPs may influence the analysis methods of click-chemistry reaction.

Response: We appreciate the reviewer’s concern regarding the separation of small nanoparticles with sizes similar to larger proteins. Actually, a more important difference is in the density of nanoparticles and proteins. Wang et al (ref 2 of manuscript) studied the protein corona on quantum dots with sizes <

10 nm and as they are denser than the proteins, they could separate the nanoparticles with centrifugation (18000 g, 20 min, the same speed as we used for our particles.). Decreasing the size of nanoparticles to the sizes in the range of protein sizes can also reduce the density and affinity of proteins to nanoparticles and their tendency to denature on nanoparticles. It should be noted that centrifugation is not a prerequisite for our technique. The key requirement for application of our approach to identify SC proteins is that it is already possible to carry out HC profiling by using any kind of separation methods (centrifugation, magnetic separation, size-based techniques, etc.). We added this discussion to the results:

PAGE.9 Although centrifugation is not a prerequisite for this method and other separation methods such as magnetic separation or size-based separation techniques can be applied on different nanoparticles. Our method is applicable to nanoparticles whose HC profiling is possible by using some kind of separation method.

We also appreciate the reviewer's concern regarding the influence of the nanoparticles (with or without metallic core) on the analysis of click-chemistry. We should mention that there is no interference from nanoparticles in the click-chemistry reaction to capture soft corona proteins; however, in the optimization process, the metallic core or particles with reducing ability can potentially interfere with the fluorescent measurement of DBCO-Sulpho-CY5 or BCA assay. For this type of nanoparticles, we developed a method to elute the corona proteins first and then quantify the proteins or azide labelling efficiency of proteins. We added the following schematic overview of the protocols developed as Supplementary.Fig.1

Supplementary Fig. S1. Schematic overview of the protocols developed for quantification of proteins and azide groups on both particles with and without interference.

We also added the following to the method part of the supplementary file:

PAGE.4 “For nanoparticles that interfere with the analysis, the proteins should be first eluted from nanoparticles and then the click reaction between DBCO-Sulpho-Cy5 be performed. We performed eluting protocol for SNPs and compared the results with a non-eluting protocol. To elute the proteins,

the nanoparticles were resuspended in 500 µl of 1 % acetic acid and incubated overnight. Then, the solution was removed by a centrifugal evaporator and the proteins in the pellet were quantified by BCA assay (which is explained in the following). Then, the proteins were incubated with DBCO-Sulpho-CY5 for the click reaction and the unreacted dyes were eliminated by using a Sephadex G-25 in PD-10 Desalting Columns (GE Healthcare Life Sciences).”

PAGE.5 “For nanoparticles that interfere with the BCA assay, the proteins should be eluted first from nanoparticles by the protocol that explained before in the “Azide modification of hard corona (HC) proteins” section and then analyzed by BCA assay.”

We also added the following to the supplementary results:

PAGE.34 “For nanoparticles that interfere with the BCA assay and azide quantification with DBCO-Sulpho-CY5, we developed a method to first elute the proteins from nanoparticles and then analyse with BCA and azide quantification assay (Supplementary Fig.1a). We developed this method for SNPs, but it can be applied to all other types of nanoparticles. The results of the BCA assay and click reaction showed the same efficiency for both eluting and non-eluting protocols.”

3. The formation of the corona is subject to crowding/cooperative effects. With the increasing combination of substrate and macromolecule, the binding of later ligands will be affected due to the presence of ligands. The crowding effects will lead to instability in the functionalization of HC proteins with azide.

Response: We agree with the reviewer that the the formation of the protein corona at material surfaces will be greatly influenced by crowding and has the potential to be modulated by cooperative effects. There is likely to be a significant heterogeneity in the protein orientation and position both across individual nanoparticle surfaces but also between nanoparticles. An important point raised by the reviewer is that crowding effects have the potential to limit accessibility of parts of the surface or parts of a protein to both other proteins and other molecules. This is an implicit part of the process by which the protein corona forms. We couple azides to sites on the proteins forming the hard protein corona and it is correct that for any given particle the positional distribution of these azide groups will be different. The functionalisation process will not be able to label all parts of the proteins because of limited accessibility by the bifunctional coupling molecule. However this molecule is small <1kDa compared to the proteins we aim to capture and so crowding effects should not limit the use of functionalisation with azides for capture since if the protein can reach a particular site then the linker

group will also be able to reach it. Nonetheless, the distribution of azides will be heterogeneous and we have added a sentence clearly stating that in the part where we highlight the limitations/requirements of the approach. The overall effects of the limited number of azides we choose to add and the heterogeneous relative orientation of proteins are part of the reason that we may underestimate the number of proteins binding. However, the approach of azide functionalisation is able to label homogeneously across the protein profile (as shown by coupling of a fluorophore and comparison to fluorescence staining profiles to coomassie staining profiles in the same SDS gels of functionalised HC proteins (Supplementary Fig. S2)). While the distribution of orientations of proteins in the HC may lead to an underestimation of amount of the different SC proteins, it does not apparently capture random bulk proteins. As such, it should give conservative identification of SC proteins as compared to the currently suggested approaches based on different forms of separation where background binding to substrates or ineffective separation from bulk proteins has the real risk of falsely identifying bulk proteins as SC proteins (false positives). We believe this gives our method a significant advantage in use in identifying SC proteins.